# Conformal elasticity of mechanism-based metamaterials

Michael Czajkowski [1], Corentin Coulais [2], Martin van Hecke[3,4] & D. Zeb Rocklin[1✉]

Deformations of conventional solids are described via elasticity, a classical field theory whose form is constrained by translational and rotational symmetries. However, flexible metamaterials often contain an additional approximate symmetry due to the presence of a designer soft strain pathway. Here we show that low energy deformations of designer dilational metamaterials will be governed by a scalar field theory, conformal elasticity, in which the nonuniform, nonlinear deformations observed under generic loads correspond with the well-studied—conformal—maps. We validate this approach using experiments and finite element simulations and further show that such systems obey a holographic bulk-boundary principle, which enables an analytic method to predict and control nonuniform, nonlinear deformations. This work both presents a unique method of precise deformation control and demonstrates a general principle in which mechanisms can generate special classes of soft deformations.

[1] School of Physics, Georgia Institute of Technology, Atlanta, Georgia 30332, USA. [2] Institute of Physics, Universiteit van Amsterdam, Science Park 904, 1098 XH Amsterdam, The Netherlands. [3] AMOLF, Science Park 104, 1098 XG Amsterdam, The Netherlands. [4] Huygens-Kamerlingh Onnes Lab, Universiteit Leiden, PObox 9504, 2300 RA Leiden, The Netherlands. ✉email: zebrocklin@gatech.edu

Mechanical metamaterials use patterns of cuts, pores and folds to achieve nonlinear[1–3], programmable[4,5], polar[6–8] and other exotic behavior[9–14] in response to external forcing. Often these features rely on the careful arrangement of the cuts, pores and folds to emulate a mechanism, which is a special pathway of deformation that enables the metamaterial to change shape at a very small (ideally zero) energy cost. For example, the canonical rotating square (RS) mechanism, which consists of perfectly hinged rigid squares, enables a uniform dilational motion (Fig. 1a) and has inspired the design of a range of metamaterials which collapse inward rather than bulge outward when compressed from the lateral direction[2,15,16]. While the dilational mechanism becomes a true zero energy motion in the limit of vanishing hinge size, for realistic metamaterials with finite hinges this uniform dilational motion is only observed for the particular case of a completely homogeneous loading condition. For generic, i.e. inhomogenous, loading conditions the response is more complex[17], and yet a general framework to describe the nonuniform soft deformations of mechanism-based metamaterials is lacking.

Here, we aim to decode the nonuniform deformations of metamaterials based on a dilational mechanism. While the elastic response of ordinary materials will locally be composed of some finite portion of shear (Fig. 1b), a dilational material will strive to expel shear everywhere in favor of the dramatically softer dilational strains. Hence, one may wonder whether the nonuniform deformations of dilational metamaterials may be locally composed of pure dilation with no shear? While not all spatial patterns of strain are realizable due to basic geometric restrictions from compatibility relations, conformal maps are well-known to constitute the full set of compatible smooth deformations which locally are composed of dilational strains only. These maps therefore provide a recipe to compose nonuniform soft modes from slow spatial variation of e.g. the RS mechanism (Fig. 1c).

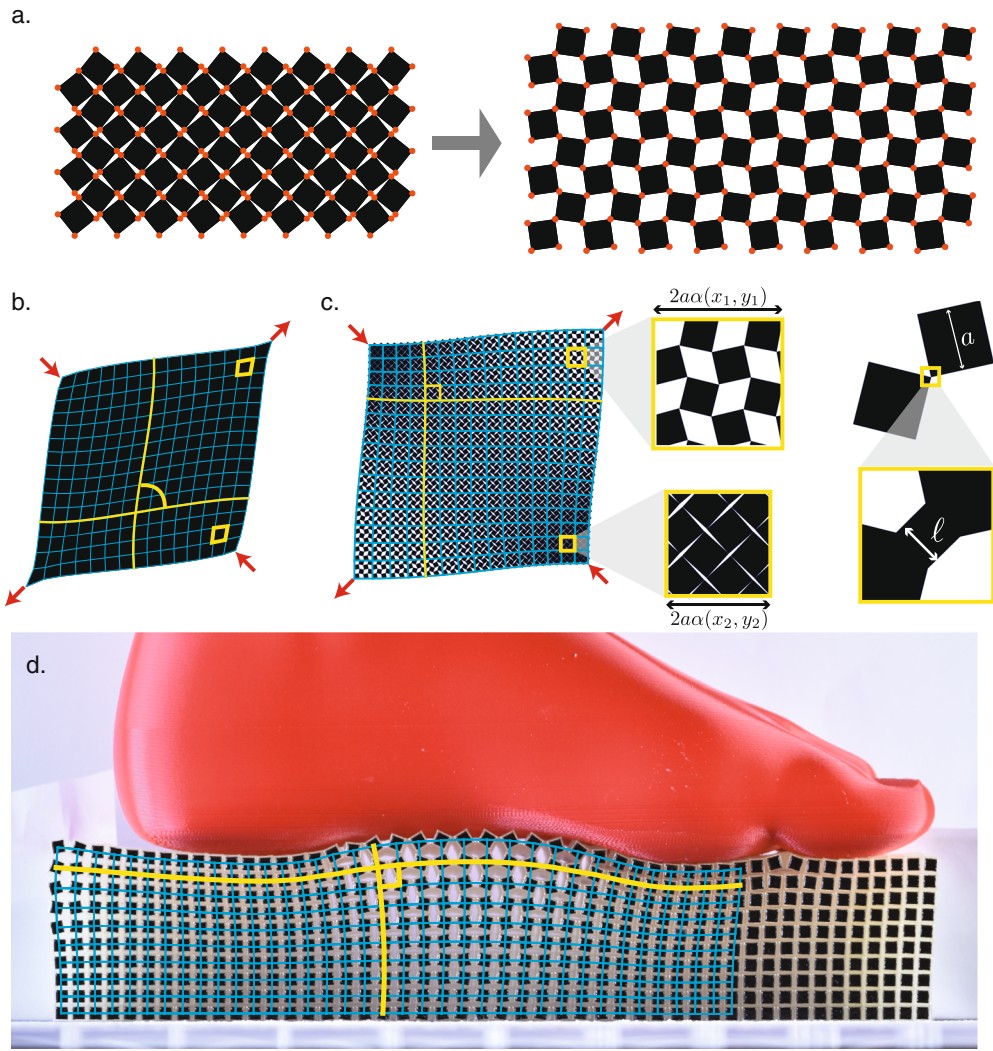

**Fig. 1 Dilational mechanism and conformal deformations. a** In the ideal rotating square mechanism, a structure of rigid squares (black) connected by frictionless hinges (red), may be dilated and contracted at zero energy cost when the squares are rotated opposite to their neighbors. **b** Applying fixed deformations (red arrows) at the boundary of a conventional elastic material leads to a spatially varying strain field that includes shear components that change the local angles between pieces of material, such as the initially perpendicular grid lines (blue, yellow). **c** In contrast, a pure dilational metamaterial designed around the RS mechanism, may accommodate the loading without shearing of the unit cells, so that even as grid lines rotate, the angles between them remain fixed. This angle-preserving behavior arises due to a local version of the RS mechanism, in which square elastic chunks of side length $a$ rotate about small, flexible hinges of thickness $\ell$ to open and close pores according to $\alpha(x, y)$, the local linear dilation factor. **d** A fabricated sample of checkerboard material approximately preserves right angles (shown in yellow) under a generic nonlinear "foot" load, suggesting conformal deformation behavior.

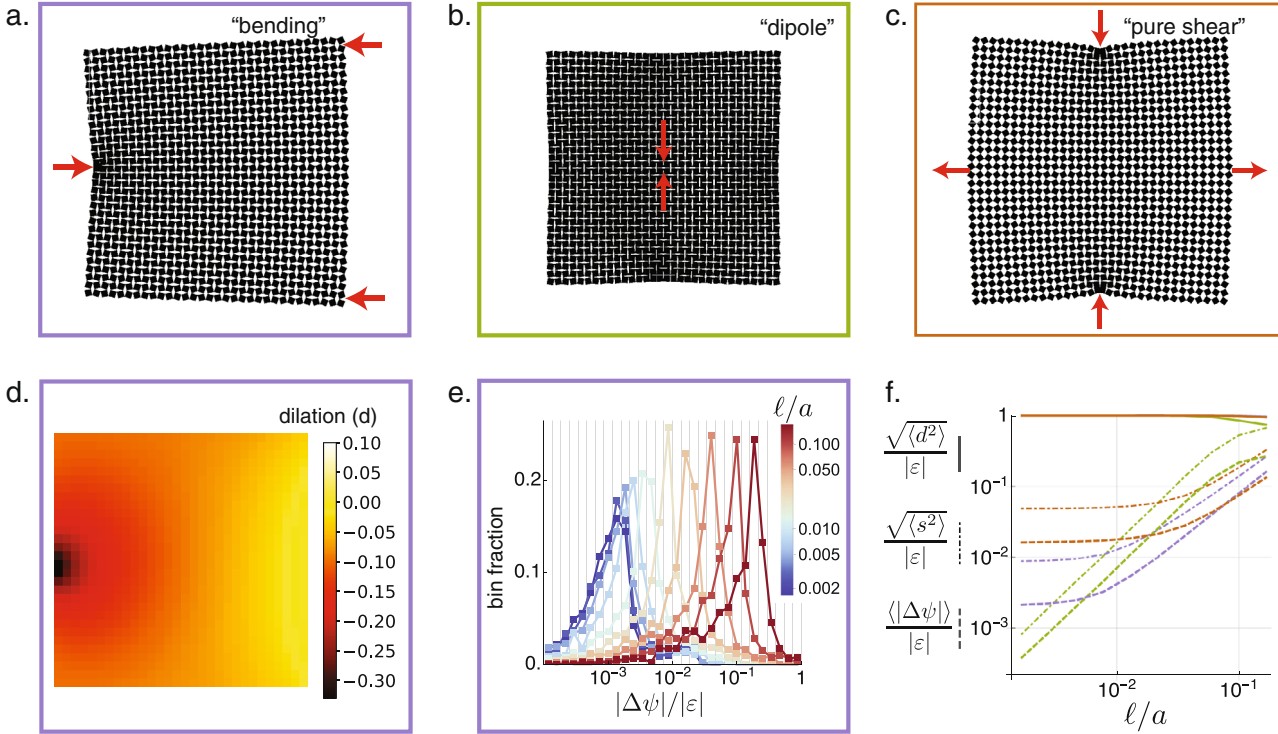

**Fig. 2 Deformations of dilational metamaterials are angle preserving. a** A finite-element simulation of a three-point bending test (the "bending" load) locates force-balanced (relaxed) states of the RS-based elastic metamaterial subject to displacements on the boundaries that are incompatible with uniform dilation. **b** The "dipole" load consists of closing a single central pore by displacement of points on opposite sides of the pore, thereby generating a localized dilation in the bulk of the material. **c** The "pure shear" load consists of compression along one axis and extension along the other, which is nevertheless compatible with a spatially varying pure dilation field. **d** The resulting dilation field under the "bending" load varies widely across the sample, including nonlinear compression and expansion. **e** Nevertheless, the angle changes $|\Delta\psi|$ remain small, even in proportion to the average strain magnitude $|\epsilon|$. Here, the histogram is shown on log-linear scale, with each square marker giving the fraction of unit cells with that angle change, and which tend to decrease as the hinge size decreases. The preservation of angles is the defining feature of conformal maps. **f** For all loads (colored in correspondence with the box outlines in (**a–c**)) the dilation fraction (solid lines) is nearly unity, and the deviation from unity is captured by the shear fraction (dot dashed lines), which for linearly small shears is the complement of the dilation fraction, and at the smallest hinge size ranges between 0.001 and 0.05. Both the shear fraction and the normalized average angle change (dashed lines) decrease with decreasing hinge size. The plateau in the bending and pure shear loads can be attributed to non-continuum lattice effects.

We therefore formulate our conformal hypothesis: under a generic and broad set of loading conditions, dilational metamaterials will respond with an angle-preserving conformal deformation (Fig. 1d), energetically much softer than conventional elastic strains.

In this work, we confirm this hypothesis using simulations, experiments, and a coarse-grained elastic theory. We first show that the response of the RS metamaterial is indeed conformal at both global and local scales. We then present a reduced elasticity theory which facilitates analytic insight, including a method for predicting nonlinear deformation. We then use the bulk-boundary correspondence principle obeyed by these metamaterials to introduce a recipe for on-demand activation of each soft configuration.

## Results

To test the hypothesis of conformal deformation, we investigate the elastic response of RS-based metamaterials at a range of hinge thicknesses using finite element (FEM) simulations which preserve the intricate pore structure (see "Methods"). While the detailed microscopic strains may contain significant shear, we use the displacement of the square centers to extract coarse-grained shear strain and dilational strain fields, shown in Supplementary Fig. 1, with the expectation that the latter should dominate as the hinges become small. Even under nonuniform strain, dilation

indeed dominates over shear in simulations of three-point "bending" (Fig. 2a), a local "dipole" dilation (Fig. 2b) and even when the system is subject to global "pure shear" via compression along one axis and expansion along the other (Fig. 2c). In these simulations, the average amount of local shear compared to dilation is captured in the shear fraction, as defined in Supplementary Note 1. Even for dilations that vary dramatically through space (Fig. 2d) this shear fraction ranges from $10^{-1.5}$ to $10^{-3}$, in contrast with values on the order of one in conventional structures (Fig. 2f). The low amounts of shear also imply nearly preserved angles, which is the defining characteristic of conformal deformations. As confirmed in Fig. 2e, f, the average angle change and the shear fraction both decrease as the hinge thickness is reduced and the system approaches the ideal mechanism limit. Despite finite-size effects from the $16 \times 16$ unit cell lattice, the strong preservation of angles indicates that the response is locally conformal.

**Fitting deformations with conformal maps.** The observation of approximately conformal local behavior suggests that deformations of the elastic RS metamaterial may also correspond globally to a conformal map. This motivates an elegant formulation expressing positions in the plane as complex numbers $\mathbf{r} = (x, y) \leftrightarrow x + iy \equiv z$ which has previously been applied to elasticity in a variety of contexts[18]. In our case, it has the added utility that

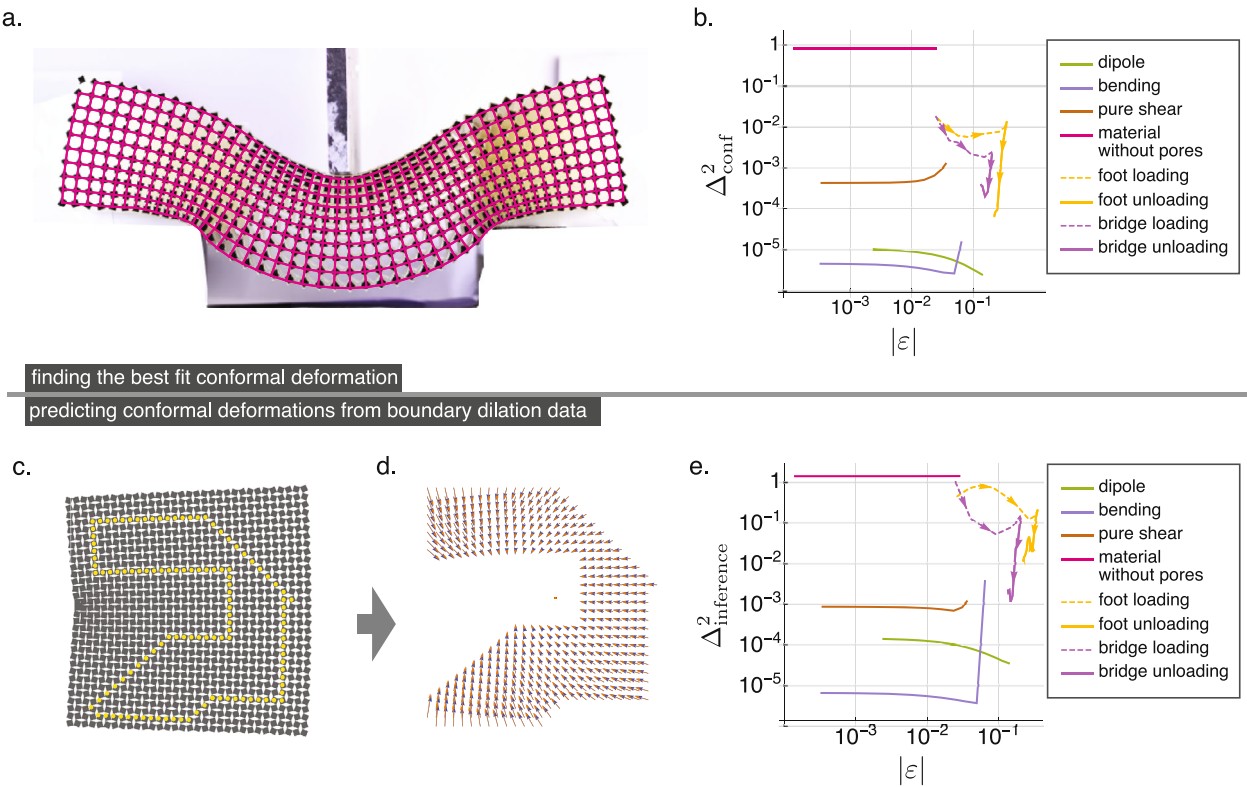

**Fig. 3 Deformations are conformal and permit nonlinear prediction. a** The nonlinear deformation of a three-point "bridge" load can be well fit to a conformal map (magenta grid) chosen to minimize the average squared deviation between the mapped and observed displacements. **b** The fraction of variance unexplained $\Delta^2_{\text{conf}}$ between observed deformations and conformal fits in both simulations and experiments is very small, demonstrating that the soft deformations correspond globally to conformal maps as well as locally to pure dilations. Experimentally, the fit is especially good in the unloading step, in which stresses arising from frictional interactions at the material boundary are able to relax. In contrast, the data for a conventional elastic material (pink) cannot be described as a conformal map. **c** According to the holographic property of conformal maps, the dilation field on the boundary of an arbitrarily chosen (gerrymandered) region, such as the one depicted in yellow for a simulation configuration, should uniquely determine the deformation within the region. **d** Indeed, the displacements inferred from the boundary dilations (orange arrows) match closely with those observed directly in simulations (blue arrows). **e** The observed displacements are now compared to the conformal maps inferred in this way, and the fraction of variance unexplained plotted as $\Delta^2_{\text{inference}}$. This demonstrates that the deformations are not merely conformal, but the precise conformal map is encoded on the boundary. Here, the predictions of experimental deformations are inferred from the full rectangular boundary, while predictions of FEM data are all inferred from the gerrymandered boundary points shown in (**c**) to show robustness to the choice of boundary shape. Note that the example chosen for display in (**c**, **d**) corresponds to the largest strain explored in the bending load, and according to (**e**) has the largest prediction error of the FEM data, and all other FEM data is predicted to even greater accuracy than depicted here.

any exact conformal deformation takes the convenient form

$$f(z) = \sum_{n=0}^{\infty} C_n z^n, \tag{1}$$

where $f(z)$ is the deformed position of the material element initially located at $z$ and the complex coefficients $\{C_n\}$ define the map. The ability to express such a map in this reduced series form, turns the search for the conformal map closest to our data into a linear algebra problem which is readily solved. Using only the first 20 $C_n$ coefficients, we use this procedure in Fig. 3a, b to identify conformal maps that are able to capture all but a small fraction $\Delta_{\text{conf}}$ (about 1%) of the observed displacements. Note here that, to appropriately assess the error in many different approximations of the observed square block displacements, we employ the fraction of variance unexplained $\Delta^2$. This measure is equivalent to the mean square displacement error normalized by the mean square magnitude of displacements; an explicit formula for this is given in Supplementary Note 2, and the diminishing returns upon increasing the number of coefficients $C_n$ is displayed in Supplementary Fig. 2. Indeed, despite the nonzero local angle deviations, we find that RS-based metamaterials will respond to loading with a deformation that is closely matched with a

conformal map even under nonlinear strains. This is in contrast with generic deformations, which fit poorly to equation (1) as shown in Fig. 3b and in general require terms with complex conjugation of $z$ such as $z^2\bar{z}^5$ included in the expansion to be captured accurately.

**Conformal elasticity**. Given this evidence that the RS metamaterial responds to loading with a near-conformal deformation, we present an elastic energy functional,

$$E = \int d^2\mathbf{r} \tfrac{1}{2}\left[G_1(\alpha)s_1^2 + G_2(\alpha)s_2^2 \right.$$
$$\left. + \tfrac{\ell^2}{a^2}M(\alpha) + a^2\tilde{M}(\alpha)|\nabla\alpha|^2\right] \tag{2}$$

which the system minimizes subject to the boundary conditions. Here, $\alpha$ is a (nonlinear) spatially varying field describing the dilation factor of the structure relative to its equilibrium, while $s_1$ and $s_2$ are the (linear) coarse pure shear and simple shear respectively, $\ell$ is the width of a hinge and $a$ the width of a square piece. $G_{1,2}(\alpha)$ are shear moduli, $M(\alpha)$ captures the dilation energy density and $\tilde{M}(\alpha)$ is the modulus associated with spatial variations in the dilation. These fields may be defined more precisely in terms of the right Cauchy–Green deformation tensor $\mathbf{C}$ (the

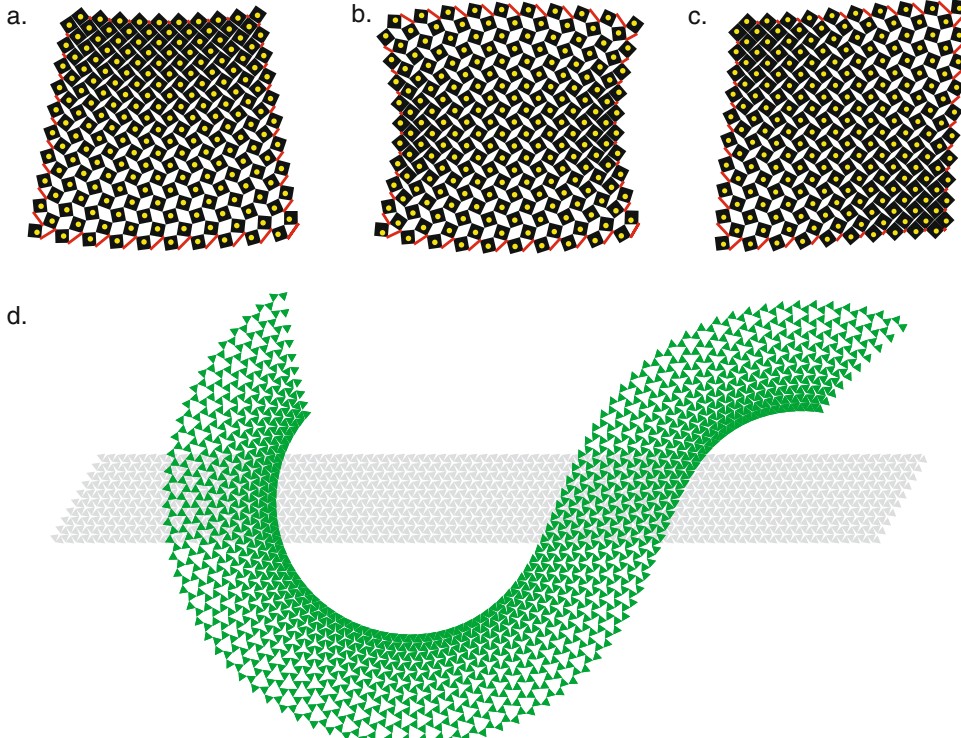

**Fig. 4 Bulk-Boundary correspondence allows for precise control of metamaterial deformation. a–c** The bulk-boundary correspondence shown in Fig. 3 enables on-demand actuation of material deformation via the addition of boundary springs (red). Here, the locations of square centers observed in simulation match well with analytically obtained target positions (yellow dots) for force-balanced configurations of a simplified ball-and-spring model. **d** Because the principles presented here extend to any dilational metamaterial, the kagome lattice may likewise be placed in a low-energy conformal deformation. The kagome lattice is capable of undergoing a broader range of deformations without self-intersection, permitting a greater range of controllable deformations such as pictured here.

metric of deformation): when we choose to orient the axes of our reference space along the vectors connecting square centers to those of their neighbors they take the form

$$\alpha = (\det[\mathbf{C}])^{1/4}, \tag{3}$$

$$s_1 = \frac{1}{2}\frac{\text{Tr}[\mathbf{C}\cdot\boldsymbol{\sigma}^{(3)}]}{\sqrt{\det[\mathbf{C}]}}, \tag{4}$$

$$s_2 = \frac{1}{2}\frac{\text{Tr}[\mathbf{C}\cdot\boldsymbol{\sigma}^{(1)}]}{\sqrt{\det[\mathbf{C}]}}, \tag{5}$$

where $\boldsymbol{\sigma}^{(1)}$ and $\boldsymbol{\sigma}^{(3)}$ are the first and third Pauli matrices. While alternate analytic approaches to the nonlinear strain of anisotropic media have been established (e.g., ref. [19]), this formalism is particularly convenient for the RS metamaterial, for which any additional terms such as couplings between dilation and shear are excluded by scaling arguments and by the the orthotropic symmetry (p4g) as presented in Supplementary Note 5. This specific form in equation (2) may also be derived by a nonlinear coarse graining, shown in Supplementary Note 4 and Supplementary Figs. 3 and 4, which provides additional useful insight into the moduli. Although other continuum elasticity theories for the RS metamaterial have been developed in one[17] and two dimensions[11,12], and another theory for a bistable dilational material[13], this nonlinear analytic form, which centers the dilational mechanism and its gradients, has not been proposed previously. While this energy function holds a rare level of insight into nonlinear deformation, it is still quite difficult to solve analytically.

To gain more analytical traction, we employ perturbation theory in the limit in which the hinges are tiny relative to the unit cell size $l/a \ll 1$ (i.e. the soft mechanism limit) and the material sample is composed of very many unit cells (i.e., the continuum limit). Here, the second two terms in equation (2) may be regarded as small perturbations, and the energy cost of deformations that include shear (first two terms of equation (2)) become prohibitively stiff by comparison. Therefore, in this limit, dilations will begin to act as a local symmetry and the conformal maps will constitute a degenerate space of ground states of the energy. Restricting our focus to this space of ground states, we recover a nonlinear notion of conformally invariant elastic theory, which was suggested by Sun et al.[20] previously for the linear deformations of the Kagome lattice, but has otherwise been viewed as "unphysical"[21] until now.

With the nonperturbative part of the energy penalizing only shear, and the set of conformal maps forming a degenerate set of ground states, it would seem that an infinite number of conformal maps are equally likely to arise in response to generic loading such as that in the FEM simulations and experiments of Fig. 2. However, this is not the case, as the degeneracy is broken by the perturbative part of the energy functional

$$\Delta E \equiv \int \mathrm{d}^2 z \frac{1}{2}\left[\frac{\ell^2}{a^2}M(|f'(z)|) + a^2\tilde{M}(|f'(z)|)|f''(z)|^2\right], \tag{6}$$

which is now expressed in terms of the conformal map $f: z \to f(z)$ describing the local dilation and rotation, with $f'(z) \equiv \alpha e^{i\phi}$ (equation (1)). While the usual problem of reducing to a constrained elasticity theory is typically done using Lagrange multipliers, conformal maps simply have $s_1 = s_2 = 0$ and $\alpha = |f'|$, allowing equation (6) to be obtained easily as the

conformal limit of equation (2). The alternate route, utilizing Lagrange multipliers, is explored in Supplementary Note 8, and yields useful information about stress. The energy in equation (6) arises purely from the last two terms in equation (2) and simultaneously breaks the conformal invariance and the ground state degeneracy, allowing predictions of specific conformal response to be generated; we refer to this procedure as "conformal elasticity". With this, we have reduced the difficult tensorial problem of conventional nonlinear elasticity of a material with pores down to a scalar theory which is much more analytically tractable. For small loading, minimizing the energy and thus predicting the deformation is reduced to a linear algebra problem, which is readily solved. As shown in Supplementary Note 7 and Supplementary Fig. 5, these predictions closely match the observed finite-element displacements with error $\Delta^2 \approx 0.0001-0.01$ for hinges with $\ell/a = 0.005$. This result showcases both the accuracy of conformal elasticity and the mathematical convenience of conformal maps.

**Bulk-boundary correspondence generates accurate nonlinear analytic predictions**. Solving equation (6) analytically is only possible in the linear regime and relies on prior knowledge of the effective stiffnesses $M$ and $\tilde{M}$. However, as shown in Fig. 3a, b, the conformal property itself extends to nonlinear deformations. We therefore devise a scheme to extend analytic predictions to the nonlinear regime, relying on a mathematical property of conformal maps: bulk-boundary correspondence. Within our formalism, this bulk-boundary correspondence works as follows: if we know the amount of dilation along the entire boundary of a section of RS material, we can predict analytically the local dilation everywhere in the material interior from the unique conformal map derivative $f'(z)$ consistent with the boundary conditions and can further integrate to infer the displacements. We illustrate the principle in Fig. 3c–e. We retrieve the dilation on an arbitrarily chosen yellow domain in Fig. 3c from the FEM data, use the bulk-boundary correspondence to calculate the deformation inside this domain (See displacement field depicted by blue arrows in Fig. 3d), and show that these are in excellent agreement with the observed displacement data (See displacement field depicted by orange arrows in Fig. 3d). This method is robust to the arbitrary choice of boundary shape and captures >99% of block displacements even at large strains (Fig. 3e).

**Bulk-boundary correspondence generates a method for precise deformation control**. The bulk-boundary correspondence can also be used for prescribing on-demand deformations. Given a target deformation field of some section of material, we can determine the suitable actuation pattern by simply examining the local target dilation at the material boundary. We illustrate this with three different actuation patterns in Fig. 4a–c. Within a simulation described in Supplementary Note 10 and Supplementary Fig. 6 and shown in Supplementary Movie 3, we actuate the material at its boundary (the red bars represent actuators) and observe that the numerically force-balanced deformation is in good agreement with the yellow dots that mark the target centers of the squares. Of course in addition to being conformal, this procedure is only able to actuate deformations that do not stretch or compress the material beyond the physical bounds of the mechanism itself. This issue is aided by the maximum modulus principle of conformal maps, in which both the maximum and minimum dilations must occur on the material boundary. We can therefore guarantee that for any choice of actuation that varies slowly along the boundary, there is a physically valid soft conformal deformation that will be activated in the interior. We propose that such boundary-control may be achieved via the

insertion of struts in the pores near the boundary, which, with remote control actuation, would allow these soft modes to be activated remotely on-demand. Exploration of this possibility, with applications in the development of soft robotics, is reserved for future work.

While the deformations of Fig. 4 arise from relatively simple functions for the input boundary dilation patterns, each analytic form of the interior deformation turns out to be much more complicated. However, one limit is simple enough to offer intuitive insight. When the length scales at which the dilation varies are much shorter than those at which the boundary curves, the boundary may be treated as that of a semi-infinite plane. For conformal metamaterial occupying the infinite upper half plane ($y > 0$), and the dilations along the boundary ($x$-axis) constrained to satisfy $\alpha(x, y = 0) = \alpha_0(x)$, there is again only one allowed a pattern of dilation, which takes the form

$$\alpha(x, y) = \exp\left[\int_{-\infty}^{\infty} dk\, e^{ikx - |k|y} w(k)\right], \quad (7)$$

where $w(k)$ is the Fourier transform of the function $w(x) = \ln(\alpha_0(x))$. In this form, we can see that the dilation (or, rather, the logarithm of the dilation) decays into the bulk exponentially according to the lengthscale of variation of the boundary dilation. For instance, when $w(x) = w_0\cos(k_0 x)$, we will have $\alpha(x, y) = \exp\left[w_0 e^{-k_0 y} \cos(k_0 x)\right]$.

## Discussion

The intuitive concept of soft modes that are locally composed of a spatially varying mechanism is not confined to the RS mechanism explored here. The conformal modes explored here arise because they are the Nambu-Goldstone modes associated with the local dilational symmetry, rather than from the details of the specific microstructure. We therefore propose the analogy: as isometries are to thin elastic sheets[22], so are conformal deformations to dilational metamaterials. Consequently, dilational materials derived from a variety of different metastructures[20,23–26] including fractal RS structures[25,26], disordered "pruned" networks[27], nanoscale[28], and three dimensional generalizations[3,29,30] will similarly have mechanical response controlled by a set of conformal soft modes. These alternate architectures provide the possibility for greater ranges of dilation[24], opening the door for even more dramatic soft deformations such as pictured for the kagome lattice in Fig. 4d.

Note, by example, that many possible loads applied to thin sheets (e.g. applying an in-plane stretch to a flat sheet) are incompatible with isometries, and an isometric soft mode theory must break down there. Similarly, a variety of overly strict loads will be incompatible with conformal deformation and our conformal elasticity theory must similarly break down. For instance, this will happen when attempting to constrain the displacements, rather than dilations all along a closed boundary. Conveniently, we have identified two particular scenarios with either a finite number of sufficiently spaced point displacements, or with dilations controlled along a closed boundary, each of which contain a broad variety of loading possibilities which are guaranteed to be compatible with a conformal map, and are therefore governed by our elasticity theory. Identifying theories that apply beyond the soft conformal response will provide an interesting avenue for future work.

We suggest that a broad class of generic mechanisms not confined to pure dilation will also generate families of soft modes that govern material response, as were indeed observed in so-called "kirigami" structures[31] and that this may become a fundamental principle for mechanism-based metamaterials, with potential applications from footwear to soft robotics. In addition, topologically polarized systems[6,32], which have additional mathematical structure controlling an additional set of exotic boundary modes,

necessarily break the symmetry of the mechanism strain pathway that allows conformal modes and the notion of mechanism compatibility may be extended to their nondilational mechanisms while incorporating topological notions. Exploring how these new classes of soft modes may obey, e.g., a generalized bulk-boundary correspondence may yield fruitful connections to exotic field theories[33] as well as black hole and string theories where holographic principles already play a vital role.

## Methods

**Generating deformations analytically.** We analytically generate the deformations of two-dimensional metamaterials that locally resemble the uniform mechanism and a rotation and hence eliminate the dominant contributions to the elastic energy, taking advantage of a map between the plane of deformation and the complex plane. In general, a deformation map $\mathbf{x} \rightarrow \mathbf{X}(\mathbf{x})$ has deformation tensor $\mathbf{F}_{ij} \equiv \frac{\partial X_i}{\partial x_j}$ which deforms infinitesimal material vectors as $dx_i \rightarrow \mathbf{F}_{ij} dx_j$. To correspond locally to a rotation of the mechanism field, the deformation tensor must be proportionate to a rotation matrix. We seek solutions in which the rotation angle $\phi$ and the dilation factor $\alpha$ vary over space, so that $\mathbf{F} = \alpha \mathbf{R}(\phi)$. However, this generates nontrivial restrictions because of the requirement that the deformed state not tear, i.e. that $\oint d\mathbf{X} = 0$ along any path. By applying Stokes' theorem, this leads to the standard geometric or kinematic compatibility conditions:

$$\partial_1 F_{12} - \partial_2 F_{11} = 0, \tag{8}$$

$$\partial_1 F_{22} - \partial_2 F_{21} = 0. \tag{9}$$

This generates two independent conditions on the two fields, suggesting a unique solution subject to certain boundary conditions. However, it becomes convenient to define new variables $z = x + iy, \bar{z} = x - iy$, with $i$ the imaginary unit. Enforcing the conditions $\partial_z z = \partial_{\bar{z}} \bar{z} = 1, \partial_z \bar{z} = \partial_{\bar{z}} z = 0$ then requires that $\partial_z = (1/2)(\partial_x - i\partial_y), \partial_{\bar{z}} = (1/2)(\partial_x + i\partial_y)$.

By summing the second compatibility condition with $(-i)$ times the first, we find, following some algebra, that compatibility is equivalent to the complex condition $\partial_{\bar{z}}[\alpha \exp(i\phi)] = 0$. Thus, the locally dilational deformations of the metamaterial are exactly the conformal maps of the plane, which may be expressed as analytic functions $f'(z)$ whose magnitude is the local dilation and whose argument is the local orientation. Displacements can be obtained via complex integration: $f(z) = \int dz f'(z)$. This method hints at a more general class of spatially varying deformations in unimode materials, but the results in this case follow from the fact that the shear-free deformations of the structure are precisely the angle-preserving ones well-known to be described by complex analytic functions.

**Finite-element simulation protocol.** For the 2D finite-element simulations, we use the commercial software Abaqus/Standard. We use a neo-Hookean energy density as a material model, a shear modulus $\mu = 0.333$ MPa*m, bulk modulus $K_0 = 16.7$ GPa*m and plane strain conditions with hybrid quadratic triangular elements (Abaqus type CPE6H). We construct the mesh so that the thinnest parts of the samples are at least two elements across. We perform two types of simulations (i) on the metamaterials unit cells with periodic boundary conditions, on which we perform a low strain static test ($5 \times 10^{-6}$)) and (ii) on the full structure, on which we perform static nonlinear analysis.

In the simulations (i), to implement periodic boundary conditions, we define constraints on the displacements of all of the nodes at the horizontal and vertical boundaries of the unit cell and impose displacement using virtual nodes[34]. We perform four types of simulations to extract the macroscopic linear bulk modulus $K = \frac{\ell^2 M''(\alpha=1)}{4a^2}$, and shear moduli $G_1(\alpha=1)$, $G_2(\alpha=1)$. In order to extract $K$ and $G_1$, we apply biaxial compression $\varepsilon_{11}^{comp} = \varepsilon_{22}^{comp} = 5 \times 10^{-6}$, $\varepsilon_{12} = 0$ and pure shear $\varepsilon_{11}^{shear} = -\varepsilon_{22}^{shear} = 5 \times 10^{-6}$, $\varepsilon_{12} = 0$ to a unit cell with periodic boundary conditions. $K$ and $G_1$ are each extracted as a slope in linear fitting $\sigma_{11}^{comp} = 2K\varepsilon_{11}^{comp}$ and $\sigma_{11}^{shear} = 2G_1\varepsilon_{11}^{shear}$. $G_2$ is extracted using the identical procedure as for $G_1$ applied to a unit cell that is rotated by $\pi/4$. We examine a unit cell with $a = 12$mm, pretwist $\theta_0 = 0.39$ and hinge thickness $\ell = 0.1$ mm, finding $(K, G_1, G_2) \sim (35, 5.2 \times 10^4, 2 \times 10^4)$Pa*m. These modulus measurements are essential for obtaining the perturbative energy minimizing prediction of deformation, summarized briefly in the main text and given in more detail in Supplementary Note 7.

In the simulations (ii), we use three kinds of boundary conditions, dipole, bending and pure shear, by imposing displacement of specific nodes, as described in Fig. 2. We extract the coordinates (position, angle) of all the squares and use these to compute the dilation and shear field (method described in Supplementary Note 1) and to calculate the relative amount of shear, as well as the accuracy of conformal map fits and of the analytic bulk-boundary correspondence (Fig. 3). Simulations are performed for a series of strains at identical material geometry as (i) and separately at linearly small strain across a range of hinge thicknesses $\ell = \{0.02, 0.033, 0.055, 0.092, 0.155, 0.258, 0.431, 0.719, 1.199, 2.0\}$ mm.

**Experimental protocol.** We 3D printed an elongated meta-beam of length 306 mm, width 64 mm and height 40 mm, consisting of a checkerboard of lattice of $48 \times 10$ squares of side $a = 4.8$ mm connected by hinges of thickness $\ell = 0.2$ mm using a Connex 500 Stratasys 3D printer and the proprietary Stratasys Agilus 30 material (30 Shore A). This material is viscoelastic, its Young's modulus at short times is $E = 3.3$ MPa and $E = 0.6$ MPa at long times (see, e.g., Dykstra et al.[35] for a calibration). For imaging purposes, we 3D printed black square-shaped pads (3.6mm) on one edge of the sample (proprietary Stratasys material: Vero Black).

The sample was tested under two sets of nonuniform boundary conditions, (i) by a foot-shaped indenter, as depicted in Supplementary Movie 1, which was 3D printed in ABS Materials using a Dimension 3D printer (Stratasys); (ii) a three points bending (i.e. bridge) test as depicted in Supplementary Movie 2, using a universal uniaxial mechanical testing machine (Instron 5943). In parallel to the mechanical loading, we recorded frames using a high-resolution camera ($6000 \times 4000$ pixels (Nikon D5600) with a telephoto lens (Nikon 200 mm, F4), positioned at 4 m from the sample. To ensure uniform lighting, the sample was illuminated by two led panels (Bresser, LG 900 54W) and white paper was used to ensure a uniform bright background. The 3D printed material tends to highly frictional and even adhesive, therefore, we used fine powder to lubricate the interaction with the bottom surface and the foot for (i). For (i), we performed a compression-decompression cycle at a rate of 0.1 mm/s up to a maximum 20 mm stroke, during which 400 frames were recorded (1 frame/s). For (ii), we performed a compression-decompression cycle at a rate of 0.2 mm/s up to a maximum 40 mm stroke, during which 400 frames were recorded (1 frame/s). The images were processed using standard image-tesselation and tracking techniques to extract the positions of the squares with subpixel detection accuracy of 0.02 pix (1 $\mu$m). We attribute the strongly different behavior observed between compression and decompression, to a combination of frictional, viscoelastic and self-adhesion effects.

**Extracting dilation and shear strains.** To extract nonlinear measures of coarse dilation and shear strain magnitudes from the deformation of RS-based structures, we track the square center displacements. Using the vectors connecting square centers to neighboring square centers as the infinitesimal material vectors in the reference ($d\mathbf{x}^{(1)}, d\mathbf{x}^{(2)}$) and target spaces ($d\mathbf{X}^{(1)}, d\mathbf{X}^{(2)}$), we may infer the deformation tensor $\mathbf{F}_{ij} = \frac{\partial X_i}{\partial x_j}$. As the reference $d\mathbf{x}^{(1)}$ and $d\mathbf{x}^{(2)}$ are orthogonal, the recipe is simply

$$\mathbf{F} = \begin{bmatrix} \frac{d\mathbf{X}^{(1)} \cdot d\mathbf{x}^{(1)}}{|d\mathbf{x}^{(1)}|^2} & \frac{d\mathbf{X}^{(2)} \cdot d\mathbf{x}^{(1)}}{|d\mathbf{x}^{(2)}|^2} \\ \frac{d\mathbf{X}^{(1)} \cdot d\mathbf{x}^{(2)}}{|d\mathbf{x}^{(1)}|^2} & \frac{d\mathbf{X}^{(2)} \cdot d\mathbf{x}^{(2)}}{|d\mathbf{x}^{(2)}|^2} \end{bmatrix}. \tag{10}$$

Then the induced metric of deformation is obtained via $\mathbf{C} = \mathbf{F}^T \cdot \mathbf{F}$. Nonlinear dilation and shear strain magnitudes may be obtained from the invariants of the metric tensor via

$$d = \sqrt{\text{Det}[\mathbf{C}]} - 1,$$
$$s^2 = \text{Tr}[\mathbf{C}] - 2\sqrt{\text{Det}[\mathbf{C}]}. \tag{11}$$

These recipes, which are derived in greater detail in Supplementary Note 1, give the dilation as the local area change, while the shear comes from the nonconformal part of the deformation as a complex map $h : z \rightarrow h(z, \bar{z})$, satisfying $s^2 = 4|\partial_{\bar{z}} h|^2$.

**Fitting displacement data to the closest conformal map.** To identify the conformal map $f(z)$ which most closely maps a set of $n_p$ points $\{z_i\}$ (expressed in complex form) to final positions $\{z_i'\}$, we minimize the error

$$\text{Err} \equiv \sum_{i=1}^{n_p} |z_i' - f(z_i)|^2. \tag{12}$$

Utilizing the analytic expansion of $f$ (equation (1)), we may find the set of coefficients $\{C_n\}$ which minimize the error by setting the partial derivatives of the error to zero, yielding equations

$$\sum_{n=1}^{n_c} A_{mn} C_n = \sum_{i=1}^{n_p} z_i' \bar{z}_i^m, \tag{13}$$

where we have defined the matrix

$$A_{mn} = \sum_{i=1}^{n_p} \bar{z}_i^m z_i^n \tag{14}$$

and the coefficients are now cut off at a maximum $n_c$. This effectively reduces the least-squares error analysis to a straightforward linear algebra problem, which is readily solved. We note that the cutoff $n_c$ should be chosen to be much less than the $n_p$ data points, to avoid overfitting.

To evaluate the accuracy of the fit, we define a similar function to the error in equation (12)

$$\Delta^2[u(z)] \equiv \frac{\sum_{i=1}^{n_p} |u_i - u(z_i)|^2}{\sum_{i=1}^{n_p} |u_i|^2}, \tag{15}$$

where $u_i = f_i - z_i$ is the observed displacement data, $u(z) \equiv f(z) - z$ are the displacements proposed to fit the data. The functional $\Delta^2[ ]$, quantifies the "fraction of unexplained variance" in the displacements, and constitutes a general method to

quantify the amount of the deformation captured by a candidate conformal function, used in Fig. 3b, e as well as in the Supplementary Notes.

**Inferring the interior conformal deformation from boundary dilation data.** In our bulk-boundary method for predicting deformation, we are able to infer the conformal deformation $f: z \to f(z)$ which will arise from a discrete set of $M$ applied dilations $\{\alpha_k\}$ at a corresponding set of locations $\{z_k\}$ along the boundary of a simply-connected planar domain. Here, $\alpha_k = A'_k/A_k$ is ratio of local area elements before $A_k$ and after $A'_k$ deformation. The inference takes place in two steps: first an intermediate function $g(z) = \log(f'(z))$ is obtained, then this function is used to integrate for the displacements.

Because $f(z)$ will be conformal, so is the function $g(z) = \log(f'(z))$ and therefore admits a similar expansion

$$g(z) = \sum_{n=0}^{N-1} D_n z^n . \tag{16}$$

where the cutoff $N$ should be significantly less than half the number of boundary points $M$ in order to avoid overfitting. Note now that since $g(z) = \log(\alpha) + i\phi$, the real part of $g$ alone determines the dilation field. We may therefore infer the $g$ that fits the boundary conditions by minimizing the error

$$\text{err} = \sum_k^M (\text{Re}[g(z_k)] - \ln(\alpha_k))^2 . \tag{17}$$

Inserting equation (16), we minimize this error with respect to the coefficients $D_n = A_n + iB_n$ yielding the equations

$$0 = \frac{\partial[\text{err}]}{\partial A_l} = f_l^{(A)} + \sum_n^N A_n Q_{ln} + \sum_n^N B_n R_{ln}$$
$$0 = \frac{\partial[\text{err}]}{\partial B_l} = f_l^{(B)} + \sum_n^N A_n S_{ln} + \sum_n^N B_n T_{ln} , \tag{18}$$

where

$$Q_{ln} = \sum_k^M \left( r_k^{n+l} \cos(n\theta_k) \cos(l\theta_k) \right)$$
$$R_{ln} = -\sum_k^M \left( r_k^{n+l} \sin(n\theta_k) \cos(l\theta_k) \right)$$
$$S_{ln} = -\sum_k^M \left( r_k^{n+l} \cos(n\theta_k) \sin(l\theta_k) \right)$$
$$T_{ln} = \sum_k^M \left( r_k^{n+l} \sin(n\theta_k) \sin(l\theta_k) \right) \tag{19}$$
$$f_l^{(A)} = -\sum_k^M \left( \ln(\alpha_k) r_k^l \cos(l\theta_k) \right)$$
$$f_l^{(B)} = \sum_k^M \left( \ln(\alpha_k) r_k^l \sin(l\theta_k) \right) ,$$

and we have expressed $z_k = r_k e^{i\theta_k}$ in a complex polar form. equation (18) reduces the inference of $g$ to a linear algebra problem which may be readily solved using built-in tools in, e.g., Mathematica. Noted also in Supplementary Note 9, the row and column corresponding to $B_0 = \phi_0$ (the undetermined global rotation) are zero throughoutand this row-column pair must be removed before numerically solving.

As described in Supplementary Note 9, the function $g(z)$, specified by the coefficients $\{D_n\}$, is used to reconstruct the $z$-derivative of the conformal map via $\partial_z f = \text{Exp}(g)$. Generating the predictions of block displacements $u(z) = f(z) - z$, shown in main text Fig. 3d and evaluated in Fig. 3e, is a matter of complex integration, which we accomplish using the built-in numerical integrators in Mathematica.

## Data availability

The experimental and finite element simulation data generated in this study have been deposited in the Zenodo database under accession code https://doi.org/10.5281/zenodo.4646672.

## Code availability

All the codes supporting this study have been deposited in the Zenodo database under accession code https://doi.org/10.5281/zenodo.4646672.

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

## Acknowledgements

The authors thank E. Matsumoto and M. Dimitriyev for insightful discussions and suggestions, S. Koot and D. Giesen for technical assistance. C.C. acknowledges funding from the European Research Council via the Grant ERC-StG-Coulais-852587-Extr3Me.

## Author contributions

M.D.C., C.C., M.v.H., and D.Z.R. together designed research, and contributed to writing. C.C. performed experiments and FEM simulations and processed data. M.D.C. performed boundary simulations, and analyzed simulation and experimental data. M.D.C. and D.Z.R. developed analytic theory.

## Competing interests

The authors declare no competing interests.
