## [Peer Review File · Nature Communications]

Title: Conformal Elasticity of Mechanism-Based MetamaterialsREVIEWER COMMENTS

Reviewer #1 (Remarks to the Author):

In this work, the authors propose to describe the rotating square (RS) mechanism—a structure made of rigid square connected by hinges—using what they term “conformal elasticity”. The authors argue that this mechanism undergoes only volumetric deformations, and hence these deformations may be described using conformal mappings. To my knowledge, this elegant approach to describe dilatational materials is new. (Note that use of complex variables and conformal mappings in elasticity is not new, see, e.g., chapter 10 Sadd’s book “ELASTICITY Theory, Applications, and Numerics”.)

The authors suggest an elastic energy functional that describes the mechanism. Perhaps their most notable result is a bulk-boundary principle, stating that from the knowledge of the deformation of the boundary of the mechanism, they can determine the deformation inside the mechanism.

Overall, as I wrote, I like the elegant approach of the authors, and find the manuscript interesting and well-written. However, I have several issues with the work, listed next.

1. To me it seems that the functional in equation (2) is not physical, since s_1 and s_2 (which are given only in the SI) depend on the coordinate system, so evaluating it in a rotated frame will lead to a different energy. To avoid this, in nonlinear elasticity the energy of deformable solids is written in terms of the invariants of the deformation metric (e.g., the right or left Cauchy-Green tensor).
2. In the transition from equation (2) to (3), the authors argue that shear deformations become energetically unfavorable, and hence the elastic energy functional remains with only volumetric terms. When there is such an internal constraint on the deformation, the stress can no longer be determined from the deformation alone. Usually, this is treated using Lagrange multiplier. This issue is not addressed in the manuscript.
3. More generally, the study of stress is missing from the manuscript. I haven’t seen any analysis of the stress that is derived from the energy suggested by the authors.
4. Are there benchmark problems to which the authors can obtain analytical solutions using their model?
5. The authors write in the SI “To our knowledge, a parameter which quantifies what fraction of a nonlinear deformation is composed of local dilation, as opposed to local shear, has not been introduced previously.” Do the authors claim that there is no known measure of a volumetric deformation in nonlinear elasticity? I find hard to believe that this is indeed their intention, since any textbook on continuum mechanics and nonlinear elasticity (see, e.g., Abeyaratne’s lecture notes) states that $J := \det(F)$ is a measure of the volumetric deformation. (This is why models like the neo-Hookean model have the form of a shear modulus times $\text{Tr}(C)$ and a bulk modulus times $(J-1)^2$.) Can the authors clarify this?

A small comment on notation: the common convention is that the reference coordinates are denoted by

upper case and the deformed coordinates are denoted by lower case, here the authors reverse this convention.

Reviewer #2 (Remarks to the Author):

The authors report that the low-energy deformation of dilational metamaterials follows conformal map and obeys a holographic bulk-boundary principle that can guide analytical prediction for nonlinear domain deformation. In general, I found the findings in this work intrinsically a follow up from the authors' prior work on transformable topological mechanical metamaterials, in which a similar designer soft strain pathway has already been discussed [1].

[1] Rocklin, D., Zhou, S., Sun, K. et al. Transformable topological mechanical metamaterials. Nat Commun 8, 14201 (2017). <https://doi.org/10.1038/ncomms14201>

RS metamaterials seem to be an example structure that exhibits dilation dominant soft deformation. The authors discuss the bulk-boundary method, which is very interesting, but it looks to me a similar process of the domain structure design discussed in the previous work. How is the addition of boundary springs different from floppy/non-floppy mode switching? I do not want to exclude that there is merit in the basic idea of the research and in the conclusions reached, but I think the authors could do a better job in explaining why the idea that RS metamaterials would exhibit conformal elasticity is interesting. A more detailed justification of the novelty of this work is needed. Specifically, how is this finding fundamentally different from the previous work on kagome lattices.

Other comments:

1. The authors talk about "small", "tiny" shear fractions to support the observation of conformal elasticity, I wonder if there can be a more quantitative description of the conformal behavior. For example, Fig. 2e and f show that the angles are not well preserved when the hinge is thick. Shear fraction decreases with hinge thickness, which is a key parameter that determines deformation energy. In this case, is there a critical value of the hinge thickness or angle changes, below which the metamaterials can be considered to follow conformal map? If so, how is that value determined? Please provide a colorbar for the histogram plot.

2. I do not think it makes sense to state that the dilation fraction is nearly unity referring to a log scale plot (Fig. 2f).

3. In Supplemental figure Fig. 1, the local maximum shear magnitude is huge (~ 0.07). Can this be considered as locally conformal?

4. In Fig. 3d, the authors conclude on an excellent agreement between the analytical prediction and the FEM results. However, I found a clear difference between the vector fields, especially in the regions

close to the left edge. Please quantify the difference and explain.

5. Is Fig. 4 showing purely simulation results? I am having a hard time understanding the actuation scheme – what is the stiffness of the added spring? Would the spring stiffness affect the control of deformation? It might be helpful if the authors can propose ideas for potential experimental realization in their current platform.

Minor issues:

- Page 1: ... much softer than than ...
- Page 1: ... in conventional structures (Fig 2. f,g). I cannot find Fig. 2g
- Page 5: I cannot find red domain in Fig. 3c.

Reviewer #3 (Remarks to the Author):

The paper focuses on a particular class of material, dilational metamaterial. While conventional material deformation may include pure dilation and shear, the deformation of this mechanism-based metamaterial is locally purely dilatational. This exciting feature provides the incentive to try to use conformal maps to describe deformations of these materials. There are two main benefits in using conformal maps for describing the nonlinear deformations of mechanism-based metamaterials. First, it reduces the difficult tensorial problem of conventional nonlinear elasticity of mechanism-based metamaterials down to a scalar theory which is much more analytically tractable. Second, the analytical approach would help us explore the reason for the unusual behavior of these materials.

Main Results: The first step is to show that deformations of dilational metamaterials are angle preserving. For this purpose, a rotating square mechanism is numerically studied in three different loading cases: bending, dipole, and pure shear. Results indicate that the systems are predominantly dilational in all loading cases, meaning that the shear force does not play a significant role. Hence, angles are preserved and the preservation of angles is the defining feature of conformal maps. Now that we know conformal maps can describe the deformations, the next step is to find a map describing deformation from a reference space to a target space. It is the most important part of the paper that encompasses most of the main text and the whole supplementary information. Figure 3 of the main text shows that the result obtained using a conformal map is compared with experiments and simulations. It shows that the proposed conformal map can almost accurately describe the nonlinear deformation. Additionally, it is shown that The bulk-boundary correspondence can also be used to control the deformation of metamaterials. For a target deformation field of some material section, it is possible to determine the suitable actuation pattern at the material boundary. It is shown in figure 4 of the main text.

Questions:

- As shown in figure 2, the results can be accurately obtained using numerical methods (Finite element

method), so what is the main reason for using this analytical mapping? (Besides gaining more analytical traction). As stated in section two of the Supplementary Information, the least-square method is used to extract the nearest conformal map from data. Therefore, we need to perform numerical simulations or conduct experiments to find the conformal map.

- The paper states that the bulk-boundary enables on-demand actuation of material deformation. Is it a property due to the proposed model? It means that we cannot do it in numerical simulations?
- The paper argues that a tensor field theory for deformations would be complicated. Therefore, it proposes a scalar theory. Results are presented for limited loading cases and only for two lattices. Can we ensure that the method would be practical for other loading cases and other lattices?

My understanding is that this paper is more about the mathematical approach and less about the engineering application of mechanism-based metamaterials.

So it might be important to clarify this point.

- I thoroughly read the main text and sections one and two of the supplementary information. However, I have difficulty understanding some of the provided equations, such as equation 3 of the main text.

Dear Referees and Editors of *Nature Communications*,

We thank the referees for the care and insight with which they prepared their reports. We appreciate the initial positive assessments offered by the referees as well as the identification of issues to be addressed. Below, we address specific points made by the referees. In order to fully address all of the related points, we have broken some referee points into multiple pieces. We hope these extensive changes have rendered the manuscript suitable for publication in *Nature Communications*.

Note also that we have added citation near the end of the manuscript:

We suggest that a broad class of generic mechanisms not confined to pure dilation will also generate families of soft modes that govern material response, as were indeed observed in so-called “kirigami” structures [30] and that this may become a fundamental principle for mechanism-based metamaterials, with potential applications from footwear to soft robotics.

This publication “A continuum field theory for the deformations of planar kirigami” by Zheng *et al.* recently added to the arXiv extends our principal of soft modes from mechanism compatibility to non-dilational modes and illustrates the urgency and the community interest in our work.

First Referee

In this work, the authors propose to describe the rotating square (RS) mechanism—a structure made of rigid square connected by hinges—using what they term “conformal elasticity”. The authors argue that this mechanism undergoes only volumetric deformations, and hence these deformations may be described using conformal mappings. To my knowledge, this elegant approach to describe dilatational materials is new. (Note that use of complex variables and conformal mappings in elasticity is not new, see, e.g., chapter 10 Sadd’s book “ELASTICITY Theory, Applications, and Numerics”.) The authors suggest an elastic energy functional that describes the mechanism. Perhaps their most notable result is a bulk-boundary principle, stating that from the knowledge of the deformation of the boundary of the mechanism, they can determine the deformation inside the mechanism. Overall, as I wrote, I like the elegant approach of the authors, and find the manuscript interesting and well-written. However, I have several issues with the work, listed next.

We thank the referee for pointing out that complex variable approaches have been successfully applied in elasticity previously. We appreciate the referee’s reference chapter, and we have included a reference to a book-length treatment, A. H. England’s “Complex Variable Methods in Elasticity”, where first we introduce complex variables:

This motivates an elegant formulation expressing positions in the plane as complex numbers $\mathbf{r} = (x, y) \leftrightarrow x + iy \equiv z$ which has previously been applied to elasticity in a variety of contexts [18]. In our case, it has the added utility that any exact conformal deformation. . .

As with the referee, while we are aware of a number of past approaches using complex variables (in which positions in the 2D plane are given in terms of z, \bar{z}) we are unaware of approaches using conformal functions (in which the deformation becomes dependent on z only). Note also that we now use the symbol \equiv to clarify that this relationship between z and \mathbf{r} is definitional.

We are glad that the referee found our approach, which they accurately summarize, to be elegant and our manuscript interesting and well-written. We now quote and address the specific issues identified by the referee.

Point 1

To me it seems that the functional in equation (2) is not physical, since s_1 and s_2 (which are given only in the SI) depend on the coordinate system, so evaluating it in a rotated frame will lead to a different energy. To avoid this, in nonlinear elasticity the energy of deformable solids is written in terms of the invariants of the deformation metric (e.g., the right or left Cauchy-Green tensor).

We regret that our previous submission was not clear about the meaning of s_1, s_2 . However, the theory presented was entirely physical, and observes the appropriate rotational invariance properties. To address and clarify this we now include explicit formulae for them in the main text, just after Eq. (2), in terms of the right-Cauchy-Green tensor which are therefore invariant under rotations of the target space:

These fields may be defined more precisely in terms of the right Cauchy-Green deformation tensor \mathbf{C} (the metric of deformation).

When we choose to orient the axes of our reference space along the vectors connecting square centers to those of their neighbors they take the form

$$\begin{aligned}\alpha &= (\det[\mathbf{C}])^{1/4}, \\ s_1 &= \frac{\text{Tr}[\mathbf{C} \cdot \sigma^{(3)}]}{\sqrt{\det[\mathbf{C}]}} , \\ s_2 &= \frac{\text{Tr}[\mathbf{C} \cdot \sigma^{(1)}]}{\sqrt{\det[\mathbf{C}]}} ,\end{aligned}$$

where $\sigma^{(1)}$ and $\sigma^{(3)}$ are the first and third Pauli matrices.

We note, though, that rotating the reference space both can and should alter the energy: the material is orthotropic, not isotropic, and therefore shears along different directions are not energetically equivalent. For this reason, we need the entire right Cauchy-Green tensor, since its invariants alone cannot distinguish the two types of shear.

Point 2

In the transition from equation (2) to (3), the authors argue that shear deformations become energetically unfavorable, and hence the elastic energy functional remains with only volumetric terms. When there is such an internal constraint on the deformation, the stress can no longer be determined from the deformation alone. Usually, this is treated using Lagrange multiplier. This issue is not addressed in the manuscript.

Indeed, we make no mention of Lagrange multipliers in the main text of the paper, but the use of Lagrange multipliers is intriguing, particularly as it relates to stress. We have included a new section in the SI, Section VIII (and included here after the third Referee responses) and we include a reference to the result in the main text:

While the usual problem of reducing to a constrained elasticity theory is typically done using Lagrange multipliers, conformal maps simply have $s_1 = s_2 = 0$ and $\alpha = |f'|$, allowing Eq. (6) (previously Eq. (3)) to be obtained easily as the conformal limit of Eq. (2). The alternate route, utilizing Lagrange multipliers, is explored in the SI section VIII, and yields useful information about stress.

We also note that the case of conformal elasticity leads to a somewhat nonstandard level of utility for the Lagrange multipliers. A standard Lagrange multiplier problem of constrained minimization could be divided into three parts: (1) satisfying the constraints, (2) obtaining the minimum within the constrained space and (3) obtaining the generalized force that must be present to enforce the constraint. Even in a typical Lagrange multiplier context, the technique only helps with the latter two. However, such techniques are commonly necessary for locating such constrained (i.e. incompressible) equilibrium states. However, the case of conformal elasticity allows for a remarkable level of analytic insight into such equilibrium states without the need for such Lagrange multiplier methods, since it is well-known (and verified here) that the space of shear free deformations is exactly the conformal maps. For instance, consider the boundary control problem: here there is only one possible conformal map consistent with the boundary conditions, and a constrained minimization need not be performed. Therefore, the main result of the Lagrange multipliers in our case is to obtain the shear stresses that enforce the shear-free conditions. As predicted by the referee, this reveals that although deformations do result in shear stresses there are also shear stress patterns that can be present and force balanced independent of the deformation.

Point 3

More generally, the study of stress is missing from the manuscript. I havent seen any analysis of the stress that is derived from the energy suggested by the authors.

Indeed, we did not initially include a discussion of the stress tensor corresponding to our theory in Eqs.(2) or (6) (**previously Eq.(3)**) as such an investigation was not required to obtain any of our results. In addition to the discussion of stress in response to the referee's previous point regarding Lagrange multipliers, we have now derived and included a discussion in SI Section VI of the (conventional, yet nonlinear and including strain gradient terms) tensorial stress arising from the elastic theory in Eq. (2). This latter analysis is done in conventional vector coordinates; because it contemplates shear strains the utility of complex coordinates is diminished and we feel it would only complicate matters.

Point 4

Are there benchmark problems to which the authors can obtain analytical solutions using their model?

Indeed, in introducing this new technique we wish to benchmark it by applying it to familiar classes of problems. To our minds, our original submission includes a number of examples that may be regarded as analytical benchmarks. In particular, they are: the four-point deformation of Fig. 1c, the foot-shaped indentation of Fig. 1d, the three-point bending test of Fig. 2a, the dipole deformation of Fig. 2b, the pure shear boundary conditions of Fig. 2c, the bridge indentation of Fig. 3a, and the four unnamed deformations of Fig. 4. To varying degrees, these meet the “benchmark” criteria of appearing commonly in the literature, with the three-point bending test in particular appearing in a wide variety of contexts.

We obtain analytical equations to predict the resulting conformal deformations, though the geometry of the rectangular samples of RS metamaterial means that a computer is used to solve a final linear algebra equation. To highlight a particularly simple geometric limit which can be treated entirely with pen-and-paper analytical methods we now add the following text to the reference to Fig. 4 in the penultimate equation of the main text:

While the deformations of Fig. 4 arise from relatively simple functions for the input boundary dilation patterns, each analytic form of the interior deformation turns out to be much more complicated. However, one limit is simple enough to offer intuitive insight. When the lengthscales at which the dilation varies are much shorter than those at which the boundary curves, the boundary may be treated as that of a semi-infinite plane. For conformal metamaterial occupying the infinite upper half plane ($y > 0$), and the dilations along the boundary (x-axis) constrained to satisfy $\alpha(x, y = 0) = \alpha_0(x)$, there is again only one allowed pattern of dilation, which takes the form

$$\alpha(x, y) = \exp \left[\int_{-\infty}^{\infty} dk e^{ikx - |k|y} w(k) \right], \quad (1)$$

where $w(k)$ is the Fourier transform of the function $w(x) = \ln(\alpha_0(x))$. In this form, we can see that the dilation (or, rather, the logarithm of the dilation) decays into the bulk exponentially according to the lengthscale of variation of the boundary

dilation. For instance, when $w(x) = w_0 \cos(k_0 x)$, we will have $\alpha(x, y) = \exp [w_0 e^{-k_0 y} \cos(k_0 x)]$

Point 5

The authors write in the SI “To our knowledge, a parameter which quantifies what fraction of a nonlinear deformation is composed of local dilation, as opposed to local shear, has not been introduced previously.” Do the authors claim that there is no known measure of a volumetric deformation in nonlinear elasticity? I find hard to believe that this is indeed their intention, since any textbook on continuum mechanics and nonlinear elasticity (see, e.g., Abeyaratne’s lecture notes) states that $J := \det(F)$ is a measure of the volumetric deformation. (This is why models like the neo-Hookean model have the form of a shear modulus times $\text{Tr}(C)$ and a bulk modulus times $(J - 1)^2$.) Can the authors clarify this?

Indeed, as the referee surmises, it is not our intention to claim that there is no known or common measure of volumetric deformation. Our claim is rather that we are not familiar with a commonly accepted way of *comparing* the amount of the amount of dilation to the amount of shear. Consider a system subject to some uniform nonlinear strain. As the referee notes, there is a fairly natural way to describe the amount of dilation, which is in fact the one we use. The referee also seems to suggest a measure of shear, though we do not wish to use this measure, $\text{Tr}(C)$ since we seek a measure of shear that is zero for a pure dilation.

In the SI section I, after we have introduced the measures of dilation and shear, we now note

While both dilation and shear are strain quantities, there is some ambiguity in comparing them nonlinearly. The expressions used here, composed entirely in terms of invariants of \mathbf{C} , are constructed to be intuitive and geometric. In addition, rewriting

$$s^2 = (\partial_x u_x - \partial_y u_y)^2 + (\partial_x u_y + \partial_y u_x)^2 \quad (2)$$

in terms of the components of the displacement $\mathbf{u} = \mathbf{x} - \mathbf{X}$ reveals that our definition of the shear matches with the combined magnitude of pure and simple shears (first and second terms in Eq. 2 respectively) from linear strain theory. Note that, in the linear limit, these strain measures satisfy $1/2(d^2 + s^2) = \epsilon_{ij}\epsilon_{ij}$.

While alternate definitions of shear may be identified which also satisfy this in the linear limit, we have explored these and they do not qualitatively change the results presented here. Finally, for a conventional simple shear $\mathbf{u} = \gamma y \hat{x}$, our formula reassuringly recovers $s^2 = \gamma^2$ to nonlinear order.

Note that we now also acknowledge the definition of α in terms of the determinant of the right Cauchy-Green tensor in the main text; see the “Proposed main text edit” section just after the third Referee responses.

Point 6

A small comment on notation: the common convention is that the reference coordinates are denoted by upper case and the deformed coordinates are denoted by lower case, here the authors reverse this convention.

We thank the referee for pointing out this departure from a useful convention. We have adjusted our notation to match this convention.

Second Referee

The authors report that the low-energy deformation of dilational metamaterials follows conformal map and obeys a holographic bulk-boundary principle that can guide analytical prediction for nonlinear domain deformation. In general, I found the findings in this work intrinsically a follow up from the authors prior work on transformable topological mechanical metamaterials, in which a similar designer soft strain pathway has already been discussed [1]. — [1] Rocklin, D., Zhou, S., Sun, K. et al. Transformable topological mechanical metamaterials. Nat Commun 8, 14201 (2017).

<https://doi.org/10.1038/ncomms14201> — RS metamaterials seem to be an example structure that exhibits dilation dominant soft deformation. The authors discuss the bulk-boundary method, which is very interesting, but it looks to me a similar process of the domain structure design discussed in the previous work. How is the addition of boundary springs different from floppy/non-floppy mode switching? I do not want to exclude that there is merit in the basic idea of the research and in the conclusions reached, but I think the authors could do a better job in explaining why the idea that RS metamaterials would exhibit conformal elasticity is interesting. A more detailed justifica-

tion of the novelty of this work is needed. Specifically, how is this finding fundamentally different from the previous work on kagome lattices.

We thank the referee for their interest in prior work by one of the authors and his collaborators, which we will refer to here as TTMM. While both of these two works concern unimode mechanical metamaterials, there are fundamental distinctions and we in no way regard the present work as a follow-up to TTMM. TTMM is fundamentally a study of topologically polarized systems that cannot be conformal whereas the present work is a study of conformal systems that cannot be topologically polarized; this and other major differences in the systems studied, the methods used and the results obtained are given in a detailed justification below. We thus feel confident in the novelty of the present work in light of TTMM and other past works.

In order to clarify the relationship between the present work and past and potential future work on 2D unimode materials, we add the following to our main text conclusions:

In addition, topologically polarized systems [6, 31], which have additional mathematical structure controlling a separate set of exotic boundary modes, necessarily break the symmetry of the mechanism strain pathway that allows conformal modes and the notion of mechanism compatibility may be extended to their nondilational mechanisms while incorporating topological notions.

We thank the referee for keeping an open mind, and here present a detailed justification of this novelty:

TTMM System and mode: The TTMM system is a deformed kagome lattice. It is two-dimensional and mechanically critical or “Maxwell” in the sense of having balanced numbers of degrees of freedom and constraints. This guarantees that it is unimode, in the sense of having a rigid nonlinear mode that is uniform between unit cells. This mode is generically a mixture of shear and dilation that evolves (for example, it could begin as mostly shear and end as mostly dilation). Somewhat contrary to the referee’s characterization, the titular phenomenon, discussed below, occurs only in the shear-dominated rather than the dilation-dominated regime. Mechanical criticality also guarantees that it will have rigid linear modes of deformation along its open boundaries. The experimental system is realized by hand-assembled, hinged pieces.

Present system and mode: A rotating squares system with hinges of finite thickness. In the limit of perfect hinges, the system has a rigid dilational

mode. Unlike the TTMM case, this mode is governed by discrete symmetry (reflectional, rotational) rather than mechanical criticality, which also ensures that the mode remains dilational rather than evolving to develop a shear component. Because the system is above the Maxwell point, this mode is the only rigid mode and the system has no boundary rigid modes.

TTMM analytical approach: The crucial analytical tool of TTMM is the construction of a mechanically critical rigidity matrix and the calculation in reciprocal space of a topological invariant. This invariant, when it assumes nontrivial values, controls the placement of rigid modes on the boundaries, shifting them for example from the left side to the right side, which can dramatically change the stiffness of the boundary. In contrast, the present system is not mechanically critical, does not have any such invariant, and is forbidden by its symmetry from developing an excess of rigid modes on any boundary (though such modes are also precluded by the lack of mechanical criticality).

Present analytical approach: In the present work, we allow the unimode to spatially vary throughout space and impose kinematic/geometric/mechanical compatibility conditions, which are solved nonlinearly by the conformal maps. In contrast, TTMM obtained its modes primarily by examining the nullspace of the rigidity matrix and did not obtain analytic results for nonlinear, nonuniform modes. One could apply our present analysis to the TTMM system, but it would be more difficult (the equations would not be analytically solvable) and less fruitful (the modes would not have some of the advantageous mathematical properties of conformal maps). Upon identifying the low-energy (but not rigid, in contrast to the TTMM modes) modes as conformal, we then construct and verify an energy functional based on the conformal modes.

TTMM results obtained: In TTMM, it was shown that the nonlinear global mode could be used to change the topological polarization and hence the presence of rigid modes and the stiffness of boundaries and interfaces.

Conformal results obtained: In the present work, we show that low-energy modes of dilational metamaterials, particularly the rotating square system, are conformal and that this insight can be used to quickly and robustly predict the deformations of the system under a broad variety of common boundary conditions and to achieve target shapes.

Other issues: The referee notes in particular the domain structure of TTMM and a possible relationship between the boundary springs of the present work and the boundary floppy modes of TTMM. The domain struc-

ture of TTMM is imposed by crafting domains with different-shaped triangular pieces; in the present work there are no domains and all pieces have the same square shape. In the present work, the boundary springs which are present in Fig. 4 (but not in earlier parts of the paper) are externally imposed conditions used to achieve a particular lightly stressed target shape. In contrast, the floppy modes of TTMM emerge on a boundary by necessity of mechanical criticality (though their precise location is determined by the topological invariant) and represent families of energetically degenerate configurations rather than a single energy-minimizing configuration. Details of simulation and experiment/physical prototyping also differ substantially between the two works.

Point 1

The authors talk about “small”, “tiny” shear fractions to support the observation of conformal elasticity, I wonder if there can be a more quantitative description of the conformal behavior. For example, Fig. 2e and f show that the angles are not well preserved when the hinge is thick.

As the referee notes, shear becomes significant when the hinges grow thick; indeed it can dominate when the pores vanish and the material is incompressible. As the term “conformal” is defined as the property of angle preservation, we present two quantitative metrics of the conformal behavior. First is the local conformal measure, the angle changes normalized by the strain magnitude. The smaller this number, averaged over the material, the more conformal the deformation. The second measure is a global measure, quantified by the error in fitting the entire deformation to a single conformal map. Both of these quantitative metrics are displayed and discussed in the manuscript. This measure of locally conformal is shown in Fig. 2f and the measure of globally conformal is shown in Fig. 3b.

Our language is meant to indicate that conformal elasticity occurs in the small hinge limit, currently achieved in common experiments. We have updated our language in the caption of Fig. 2 to incorporate the quantitative measures of this smallness that, as the referee notes, is already present in our Fig. 2ef:

For all loads... the dilation fraction (solid lines) is nearly unity, and the deviation from unity is captured by the shear fraction

(dot dashed lines), which for linearly small shears is the complement of the dilation fraction, and at the smallest hinge size ranges between 0.001 – 0.05.

Shear fraction decreases with hinge thickness, which is a key parameter that determines deformation energy. In this case, is there a critical value of the hinge thickness or angle changes, below which the metamaterials can be considered to follow conformal map? If so, how is that value determined?

There is no critical value here, rather our theory and data both indicate that smaller hinges and finer lattice structures will give better conformal description in a continuous sense. We have derived estimates of this shear fraction as a function of hinge thickness and lattice coarseness, finding continuous quadratic rather than exponential decrease in the magnitude of the shear fraction. However, there is something like a critical (more like threshold) value of the hinge thickness at which smaller hinges stops helping reduce the shear fraction, due to finite size effects, as shown in the plateau of the curves in the plots you reference.

Please provide a color bar for the histogram plot.

We thank the referee for noting this omission in Fig. 2e. We have now included a color bar.

Point 2

I do not think it makes sense to state that the dilation fraction is nearly unity referring to a log scale plot (Fig. 2f).

Indeed, the fact that the dilation fraction appears to be unity on a log plot does not indicate its actual deviation from unity very precisely. Our intent was to communicate that the shear fraction also present on the same plot indicates the deviation of the dilation fraction from unity. We also seek to clarify that the story is slightly more complicated, insofar as this complementarity between the two measures (that is, the fact that they sum to one) is true only when the shear is small, as is the case for the situations we consider. To address this, we revise our original text for clarity as shown in the response to your point 1.

Point 3

In Supplemental figure Fig. 1, the local maximum shear magnitude is huge (0.07). Can this be considered as locally conformal?

Thank you for pointing this out. As you note, the amount of shear can sometimes become large (yet still nearly an order of magnitude smaller than the dilation amount in the same location) in regions where force is being applied and where the dilation amount is changing quickly through space. In a sense, this is where we might consider the deformation as locally “less conformal”. The importance of these plots is more to indicate that away from these small parts of the boundary where force is applied the shear fraction is still quite small, and thus the overall system shape is very much near conformal. We add language to the SI Section I to clarify this:

As shown in (SI) Fig. (1), these formulae reveal that the typical magnitude of local dilation is much larger than that of local shear. *While the shear may become large in small regions near the applied loads and where the dilation changes over short distances, dilation still dominates by nearly an order of magnitude at the worst.* Nonuniform, nonlinear deformations of the RS-based mechanism in force-balance may then indeed be composed of local dilations with comparatively very little shear.

Where the new sentence is marked here in blue.

Point 4

In Fig. 3d, the authors conclude on an excellent agreement between the analytical prediction and the FEM results. However, I found a clear difference between the vector fields, especially in the regions close to the left edge. Please quantify the difference and explain.

Of course, while the agreement is visibly quite nice, the match is indeed not perfect. The error in matching between these vector fields is quantified across a range of applied strains, FEM simulations, and experiments in Fig. 3e. The method of error estimation shown here is known as the “fraction of variance unexplained” (FVU) in regression communities, yet is also understood as the mean square error in the matching these displacement fields, normalized by the mean square magnitude of displacement. We have added clarifying language to the main text to make this more clear:

Note here that, to assess the error in many different approximations of the observed block displacements, we employ the fraction of variance unexplained Δ^2 . This measure is equivalent to the mean square displacement error normalized by the mean square magnitude of displacements; an explicit formula for this is given in the SI.

As you note, the image shown has visibly perceptible error, and for this particular configuration the FVU is ~ 0.003 meaning that the visible deviation should be on the order of $\sim 5\%$, which is roughly what we observe in the Fig 3d. This particular deformation is the maximum strain explored for the bending load, and from Fig. 3e, this is the largest FVU observed in the FEM simulation data, and note that the rest of the FEM data, where there is no error from noisy image analysis, all perform significantly better than this. We add language to the Fig. 3 caption to emphasize this point:

Note that the example chosen for display in (c,d) corresponds to the largest strain explored in the bending load, and according to (e) has the largest prediction error of the FEM data, and all other FEM data is predicted to even greater accuracy than depicted here.

Point 5

Is Fig. 4 showing purely simulation results? I am having a hard time understanding the actuation scheme—what is the stiffness of the added spring? Would the spring stiffness affect the control of deformation? It might be helpful if the authors can propose ideas for potential experimental realization in their current platform.

The images presented here are examples from simplified simulations of a bar and spring model of the lattice. However, the target positions marked by yellow dots are predictions from the analytic theory and we now add “analytically obtained” where they are referenced in the caption to clarify this.

The added boundary springs are meant to emulate rigid actuators. e.g. a metal strut. The stiffness was taken to be two orders of magnitude greater than the bulk springs to reflect this increased rigidity. We have added a small section to the SI (section X) to explain the protocol here. Experimental

realization, we speculate, could arise from such actuators, as mentioned here, and as we have now indicated more clearly in the manuscript:

We can therefore guarantee that for any choice of actuation that varies slowly along the boundary, there is a physically valid soft conformal deformation that will be activated in the interior. We propose that such boundary-control may be achieved via the insertion of struts in the pores near the boundary, which, with remote control actuation, would allow these soft modes to be activated remotely on-demand. Exploration of this possibility, with applications in development of soft robotics, is reserved for future work.

Here, the new language is indicated in blue.

Point 6

Minor Issues:

- *Page 1: much softer than than*
- *Page 1: in conventional structures (Fig 2. f,g). I cannot find Fig. 2g*
- *Page 5: I cannot find red domain in Fig. 3c.*

We thank the referee for catching these typos. We have removed the extra “than”, removed the reference to Fig. 2g (we had previously removed a Fig. 2g in favor of incorporating its information into Fig. 2f) and corrected the reference to a “red domain” to the “yellow domain” with which we had replaced it.

Third Referee

The paper focuses on a particular class of material, dilational metamaterial. While conventional material deformation may include pure dilation and shear, the deformation of this mechanism-based metamaterial is locally purely dilatational. This exciting feature provides the incentive to try to use conformal maps to describe deformations of these materials. There are two main benefits in using conformal maps for describing the nonlinear deformations of

mechanism-based metamaterials. First, it reduces the difficult tensorial problem of conventional nonlinear elasticity of mechanism-based metamaterials down to a scalar theory which is much more analytically tractable. Second, the analytical approach would help us explore the reason for the unusual behavior of these materials.

Main Results: The first step is to show that deformations of dilational metamaterials are angle preserving. For this purpose, a rotating square mechanism is numerically studied in three different loading cases: bending, dipole, and pure shear. Results indicate that the systems are predominantly dilational in all loading cases, meaning that the shear force does not play a significant role. Hence, angles are preserved and the preservation of angles is the defining feature of conformal maps. Now that we know conformal maps can describe the deformations, the next step is to find a map describing deformation from a reference space to a target space. It is the most important part of the paper that encompasses most of the main text and the whole supplementary information. Figure 3 of the main text shows that the result obtained using a conformal map is compared with experiments and simulations. It shows that the proposed conformal map can almost accurately describe the nonlinear deformation. Additionally, it is shown that The bulk-boundary correspondence can also be used to control the deformation of metamaterials. For a target deformation field of some material section, it is possible to determine the suitable actuation pattern at the material boundary. It is shown in figure 4 of the main text.

We are glad that the referee shares our excitement about the link between dilatational metamaterials and conformal maps and recognizes the benefits that can be derived therefrom. We address the referee's specific points below.

Point 1

As shown in figure 2, the results can be accurately obtained using numerical methods (Finite element method), so what is the main reason for using this analytical mapping? (Besides gaining more analytical traction). As stated in section two of the Supplementary Information, the least-square method is used to extract the nearest conformal map from data. Therefore, we need to perform numerical simulations or conduct experiments to find the conformal map.

We thank the referee for considering this important point, and appreciate the opportunity to clarify the strength of our approach. The referee is correct

that when we fit experimental and numerical data to the nearest conformal map in Fig. 3b we are merely validating our conformal theory rather than applying it to make a prediction. However, in other places, particularly Fig. 3e and Fig. 4 we are in fact *predicting* particular conformal deformations using our analytical theory and only using numerical and experimental data to validate these prediction methods. We may then safely say that within the regime of validity of our theory we may rely on the analytical conformal predictions before doing an experiment or simulation. We discuss our claims of predictive power in several places in the main text of our submission, for example in the abstract (emphasis added)

We validate this approach using experiments and finite element simulations and further show that such systems obey a holographic bulk-boundary principle, which enables an unprecedented analytic method to *predict* and control nonuniform, nonlinear deformations.

We note also that, as the referee alludes, analytic “traction” has its own merits even beyond the aforementioned prediction and control. New analytic techniques can streamline and generalize results and provide new insights and identify new connections that lay the groundwork for future testable, novel theories.

Point 2

The paper states that the bulk-boundary enables on-demand actuation of material deformation. Is it a property due to the proposed model? It means that we cannot do it in numerical simulations?

We regret that our original submission was unclear. In fact we did apply the bulk-boundary method in numerical simulations in Fig. 4 and we are confident it can be performed in experiments as well. To clarify this, we expand a discussion in the main text of Fig. 4 to now read:

Within a simulation described in SI Section X, we actuate the material at its boundary (the red bars represent actuators) and observe that the numerically force balanced deformation is in good agreement with the yellow dots that mark the target centers of the squares.

Point 3

The paper argues that a tensor field theory for deformations would be complicated. Therefore, it proposes a scalar theory. Results are presented for limited loading cases and only for two lattices. Can we ensure that the method would be practical for other loading cases and other lattices?

The extension of our results to other unimode lattices is a very important topic which we are currently exploring in related work. We have chosen to limit the main focus of our paper to the rotating square lattice in order to maintain a realistic scope to the paper, and some results, such as the orthotropy and particular nonlinear response in the energy functional are certainly limited to that case. However, the underlying property of low-energy deformations being conformal should apply to any 2D structure with an underlying rigid dilational mode. We provide evidence of this mainly in our analytical theory, which derives such modes not from the specifics of the structure but only from the uniform dilational mode in the first methods section. We also validate this by mapping out a conformal soft mode in the second lattice to which the referee alludes, the kagome, in Fig. 4d. For unimode materials in two dimensions that are not dilational, we expect spatially-varying low-energy modes of a similar form to be present, but to lack the analytical tractability of the conformal modes. We comment on this generalizability in the final paragraph of the main text.

While our results are indeed tested against a limited set of loadings, these 8 cases are chosen to be rather varied and arbitrary. Indeed, the only assumption that is required for our analytic coarse-graining of the conformal elasticity theory is that there exist a near-conformal deformation which is compatible with the boundary conditions. This existence turns out to be a bit nontrivial, and exploration of the over-constrained conditions which cause the theory to breakdown will provide an interesting avenue for ongoing future work. In analogy, the response of thin elastic sheets is well-known to be governed by the (soft) isometries of the sheet. However, even in these sheets, the theory only applies when an isometry exists to accommodate the loading. Simply taking a flat sheet and attempting to displace two points to a greater distance will not allow a isometric soft deformation, and will be outside of a reduced theory such as that of “NambuGoldstone modes and diffuse deformations in elastic shells” by Christian D. Santangelo. Similarly, in the case of conformal elasticity, if the experimenter attempts to prescribe the *displacements* all along the boundary, rather than the dilations, there will

generally not be a compatible conformal map, and a higher-energy theory beyond the soft modes explored here must be employed.

Conveniently, our conformal elasticity does allow for a variety of loading cases which are confidently compatible with a conformal deformation. The ones we have identified fall into the categories of (1) a finite number of point loads or (2) prescribed dilation along a closed boundary. We add the penultimate paragraph of the main text to emphasize this mathematical convenience:

Note, by example, that many possible loads applied to thin sheets (e.g. applying an in-plane stretch to a flat sheet) are incompatible with isometries, and an isometric soft mode theory must break down there. Similarly, a variety of overly strict loads will be incompatible with conformal deformation and our conformal elasticity theory must similarly break down. For instance, this will happen when attempting to constrain the displacements, rather than dilations all along a closed boundary. Conveniently, we have identified two particular scenarios with either a finite number of sufficiently spaced point displacements, or with dilations controlled along a closed boundary, each of which contain a broad variety of loading possibilities which are guaranteed to be compatible with a conformal map, and are therefore governed by our elasticity theory. Identifying theories that apply beyond the soft conformal response will provide an interesting avenue for future work.

Point 4

My understanding is that this paper is more about the mathematical approach and less about the engineering application of mechanism-based metamaterials. So it might be important to clarify this point.

This paper is indeed driven by an analytical mathematical insight and approach, albeit one we have applied to an experimental system currently under broad study. However, while we have not directly demonstrated any practical applications, we do believe that such may be realized at some point in the future. Exotic examples may be soft robots that swiftly achieve an infinite multitude of target shapes in response to environmental conditions. More mundanely, an understanding of conformal deformations might explain

how auxetic structures like those that grant certain athletic shoes added flexibility lead to uneven deformations and wear. To clarify our optimism about the potential application of the current study, we add to a sentence in the original final paragraph a final clause, which now reads:

We suggest that a broad class of generic mechanisms not confined to pure dilation will also generate families of soft modes that govern material response, as were indeed observed in so-called “kirigami” structures [30] and that this may become a fundamental principle for mechanism-based metamaterials, with potential applications from footwear to soft robotics.

Point 5

I thoroughly read the main text and sections one and two of the supplementary information. However, I have difficulty understanding some of the provided equations, such as equation 3 of the main text.

Thank you for pointing this out, we very much would like the paper to be clear and understandable. We have shifted the language and expanded the development of the conformal theory to make this more pedagogical. Please let us know if there are more direct confusions or places that you find hard to understand. See the edited section below:

Proposed main text revision

Note that the changes made to this section of the main text are denoted in blue.

Given this evidence that the RS metamaterial responds to loading with a near-conformal deformation, we present an elastic energy functional,

$$E = \int d^2\mathbf{r} \frac{1}{2} \left[G_1(\alpha) s_1^2 + G_2(\alpha) s_2^2 + \frac{\ell^2}{a^2} M(\alpha) + a^2 \tilde{M}(\alpha) |\nabla\alpha|^2 \right] \quad (3)$$

which the system minimizes subject to the boundary conditions. Here, α is a (nonlinear) spatially varying field describing the dilation factor of the structure relative to its equilibrium, while s_1 and s_2 are the (linear) coarse pure shear and simple shear respectively, ℓ is the width of a hinge and a the width of a square piece. $G_{1,2}(\alpha)$ are shear moduli, $M(\alpha)$ the dilation energy density and $\bar{M}(\alpha)$ the modulus associated with spatial variations in the dilation. These fields may be defined more precisely in terms of the right Cauchy-Green deformation tensor \mathbf{C} (the metric of deformation). When we choose to orient the axes of our reference space along the vectors connecting square centers to those of their neighbors they take the form

$$\alpha = (\det[\mathbf{C}])^{1/4}, \quad (4)$$

$$s_1 = \frac{\text{Tr}[\mathbf{C} \cdot \sigma^{(3)}]}{\sqrt{\det[\mathbf{C}]}} , \quad (5)$$

$$s_2 = \frac{\text{Tr}[\mathbf{C} \cdot \sigma^{(1)}]}{\sqrt{\det[\mathbf{C}]}} , \quad (6)$$

where $\sigma^{(1)}$ and $\sigma^{(3)}$ are the first and third Pauli matrices. As discussed in the SI, additional terms such as couplings between shear and dilation are excluded by scaling arguments and by the symmetries of the lattice; and this specific form may be derived by a nonlinear coarse graining, providing additional useful insight into the moduli. Although other continuum elasticity theories for the RS metamaterial have been developed in one [17] and two dimensions [11,12], and another theory for a bistable dilational material [13], this nonlinear analytic form, which centers the dilational mechanism and its gradients, has not been proposed previously. While this energy function holds a rare level of insight into nonlinear deformation, it is still quite difficult to solve analytically.

To gain more analytical traction, we employ perturbation theory in the limit in which the hinges are tiny relative to the unit cell size $l/a \ll 1$ (i.e. the soft mechanism limit) and the material sample is composed of very many unit cells (i.e., the continuum limit). Here, the second two terms in Eq. 3 may be regarded as small

perturbations, and the energy cost of deformations that include shear (first two terms of Eq. (3)) become prohibitively stiff by comparison. Therefore, in this limit, dilations will begin to act as a local symmetry and the conformal maps will constitute a degenerate space of ground states of the energy. Restricting our focus to this space of ground states, we recover a nonlinear notion of conformally invariant elastic theory, which was suggested by Sun. *et. al.* [19] previously for the linear deformations of the Kagome lattice, but has otherwise been viewed as “unphysical” [20] until now.

With the nonperturbative part of the energy penalizing only shear, and the set of conformal maps forming a degenerate set of ground states, it would seem that an infinite number of conformal maps are equally likely to arise in response to generic loading such as that in the FEM simulations and experiments of Fig. 2. However, this is not the case, as the degeneracy is broken by the perturbative part of the energy functional

$$\Delta E \equiv \int d^2z \frac{1}{2} \left[\frac{\ell^2}{a^2} M(|f'(z)|) + a^2 \tilde{M}(|f'(z)|) |f''(z)|^2 \right], \quad (7)$$

which is now expressed in terms of the conformal map $f : z \rightarrow f(z)$ describing the local dilation and rotation, with $f'(z) \equiv \alpha e^{i\phi}$ (Eq. (1)). While the usual problem of reducing to a constrained elasticity theory is typically done using Lagrange multipliers, conformal maps simply have $s_1 = s_2 = 0$ and $\alpha = |f'|$, allowing Eq. 7 to be obtained easily as the conformal limit of Eq. 3. The alternate route, utilizing Lagrange multipliers, is explored in the SI section VIII, and yields useful information about stress. The energy in Eq. (7) arises purely from the last two terms in Eq. (3) and simultaneously breaks the conformal invariance and the ground state degeneracy, allowing predictions of specific conformal response to be generated; we refer to this procedure as “conformal elasticity”. With this, we have reduced the difficult tensorial problem of conventional nonlinear elasticity of a material with pores down to a scalar theory which is much more analytically tractable. For small loading, minimizing the energy and thus predicting the deformation is reduced to a linear algebra problem,

which is readily solved. As shown in the SI, these predictions closely match the observed finite-element displacements with correlation coefficient $R^2 \approx 0.99-0.9999$ for hinges with $\ell/a = 0.005$. This result showcases both the accuracy of conformal elasticity and the mathematical convenience of conformal maps.

Proposed new SI section on Lagrange multipliers

In this Section, we employ a more conventional method of enforcing the shear-free conditions on the RS metamaterial by the introduction of Lagrange multipliers λ_1, λ_2 . This produces insight into the shear stresses enforcing this shear-free condition. For simplicity, we consider a system without gradient terms, so that the energy takes the form

$$E = \int d^2\mathbf{R} [b(J) + \lambda_1(F_{11} - F_{22}) + \lambda_2(F_{12} + F_{21})], \quad (8)$$

where F_{ij} is the deformation tensor, $J = \alpha^2$ is the determinant of its matrix form and the general dependence on the reference coordinate has been suppressed.

The conditions for equilibrium are that the energy cannot be lowered by any movement of the material from its target position, $\mathbf{r}(\mathbf{R})$. This can be expressed as a functional derivative (See, e.g., Lazar and Kirchner [5] or Saremi and Rocklin [6]):

$$\frac{\delta E}{\delta r_1(\mathbf{R}')} = \frac{\delta E}{\delta r_2(\mathbf{R}')} = 0. \quad (9)$$

These functional derivatives inherit many properties of conventional derivatives and in particular if the energy depends on the position only through an intermediate function, such as the deformation tensor, we have the chain rule:

$$\frac{\delta E}{\delta r_i(\mathbf{R}')} = \int d^2\mathbf{R}'' \frac{\delta E}{\delta F_{jk}(\mathbf{R}'')} \frac{\delta F_{jk}(\mathbf{R}'')}{\mathbf{r}(\mathbf{R}')}. \quad (10)$$

Consequently, after some algebra, we obtain our equilibrium conditions:

$$b''(J) [F_{11}\partial_1 J + F_{12}\partial_2 J] + \partial_1 \lambda_1 + \partial_2 \lambda_2 = 0, \quad (11)$$

$$b''(J) [F_{11}\partial_2 J - F_{12}\partial_1 J] - \partial_2 \lambda_1 + \partial_1 \lambda_2 = 0. \quad (12)$$

In obtaining this expression, we have already used that $F_{22} = F_{11}$, $F_{21} = -F_{12}$ (the constraints). To this, we now add our compatibility condition, as discussed in the Methods section of the main text. We find that this compatibility condition is equivalent to

$$(\partial_1 + i\partial_2)(F_{11} - iF_{12}) = 0, \quad (13)$$

or

$$\partial_{\bar{z}} f' = 0 \quad (14)$$

where

$$z \equiv x + iy, \quad (15)$$

$$\partial_{\bar{z}} = \frac{1}{2}(\partial_x + i\partial_y) \quad (16)$$

$$f' \equiv F_{11} - iF_{12}. \quad (17)$$

We now note that $J = |f'|^2$ and we can take our two original real equilibrium conditions and add the first equation to the second equation multiplied by i , and obtain the equivalent complex condition

$$b''(J)|f'^2|\bar{f}'' + \partial_{\bar{z}}\lambda = 0, \quad (18)$$

$$\lambda(z, \bar{z}) \equiv \lambda_1 + i\lambda_2. \quad (19)$$

governing the constrained equilibrium states. Thus, there are stresses that are supported even in the absence of deformation ($f'(z) = 0$): the structure supports shear stresses that are functions of \bar{z} only. That is, they are antiholomorphic, whereas the low-energy deformations are holomorphic, or conformal. In addition, there is an additional shear stress proportional to $b''(J)$ that forms when there is a nonzero bulk modulus and a spatially varying deformation tensor. Similar results arise more generally when dilation gradient terms are included, still allowing for an undetermined antiholomorphic space of shear stresses.

Proposed new SI section on stress

Having constructed the Energy in Eq. (34), we would like to know the corresponding stress tensor, as this is the more common quantity used in nonlinear elasticity. However, this theory is both nonlinear and includes strain gradient terms, and we require a relation which appropriately incorporates these effects. To do this, we start with the essential information that the functional derivative with respect to displacement gives the force density as

$$f_i = -\frac{\delta E}{\delta u_i}. \quad (20)$$

In addition, we know that the force density is the divergence of a stress tensor

$$f_i = \partial_j N_{ji}, \quad (21)$$

and so the process of deriving the stress is a matter of taking the functional derivative in Eq. 20 and fitting it into the form of Eq. 21. Our energy functional is an integral over an energy density which is a function of our quantities α and s_1 and s_2 . These are strain quantities, and may be expressed in terms of the Lagrangian strain

$$\varepsilon_{ij} = \frac{1}{2}(\partial_i u_j + \partial_j u_i + \partial_i u_k \partial_j u_k) = \frac{1}{2}(C_{ij} - \delta_{ij}), \quad (22)$$

where $C_{ij} = F_{ki}F_{kj}$ is the right Cauchy-Green deformation tensor (i.e. the metric of deformation) and $F_{ij} = \partial_j u_i$ is the deformation gradient tensor. The variables in our energy functional Eq. (34) may be written, to lowest order in s_1, s_2 , in terms of this strain via

$$\alpha = \sqrt{\det[F]} = (\det[C])^{1/4} = (\det[2\varepsilon + \mathbb{1}])^{1/4} \quad (23)$$

$$s_1^2 = \frac{1}{2} \frac{\text{Tr}[C \cdot \sigma^{(3)}]}{\sqrt{\det[C]}} = \frac{\text{Tr}[\varepsilon \cdot \sigma^{(3)}]}{\sqrt{\det[2\varepsilon - \mathbb{1}]}} \quad (24)$$

$$s_2^2 = \frac{1}{2} \frac{\text{Tr}[C \cdot \sigma^{(1)}]}{\sqrt{\det[C]}} = \frac{\text{Tr}[\varepsilon \cdot \sigma^{(1)}]}{\sqrt{\det[2\varepsilon + \mathbb{1}]}}. \quad (25)$$

Therefore, we should think of our energy functional as having the form

$$E = \int d^d x \Phi(\varepsilon_{ij}, \partial_k \varepsilon_{ij}) \quad (26)$$

With this, taking the functional derivative in Eq. 20 is a matter of using the chain rule for functional derivatives

$$\frac{\delta E}{\delta u_l(x)} = \int d^d x' \frac{\delta E}{\delta \varepsilon_{ij}(x')} \frac{\delta \varepsilon_{ij}(x')}{\delta u_l(x)}. \quad (27)$$

Here, ε_{jk} may be thought of as a functional of u_i and this derivative is taken using the definition of the functional derivative

$$\frac{\delta F[u]}{\delta u(x)} = \lim_{h \rightarrow 0} \frac{F[u(x') + h\delta(x - x')] - F[u(x')]}{h} \quad (28)$$

which becomes

$$\begin{aligned} \frac{\delta \varepsilon_{ij}[u(x')]}{\delta u_l(x)} &= \lim_{h \rightarrow 0} \frac{\varepsilon_{ij}[u(x') + h\hat{e}_l\delta(x - x')] - \varepsilon_{ij}[u(x')]}{h} \\ &= \frac{\partial \varepsilon_{ij}(x)}{\partial(\partial_k u_l(x))} \partial_k \delta(x - x'). \end{aligned} \quad (29)$$

And using the form of Eq. 22, we find

$$\frac{\delta \varepsilon_{ij}[u(x')]}{\delta u_l(x)} = \frac{1}{2} (\delta_{lj} \partial_i + \delta_{il} \partial_j + \partial_i u_l \partial_j + \partial_j u_l \partial_i) \delta(x - x'). \quad (30)$$

Plugging this back into Eq. 27, and simplifying a bit, we may find

$$N_{ij} = \frac{\delta E}{\delta \varepsilon_{ik}} F_{kj}. \quad (31)$$

where

$$\frac{\delta E}{\delta \varepsilon_{ij}} = \frac{\partial \Phi}{\partial \varepsilon_{ij}} - \partial_k \frac{\partial \Phi}{\partial(\partial_k \varepsilon_{ij})} \quad (32)$$

is the standard functional derivative. As the divergence of this stress in terms of the reference space coordinates gives the force density in *reference* space, this construction captures the Nominal stress (also known as Engineering stress and corresponding to the transpose of the first Piola-Kirchhoff stress) in accordance with

the nonlinear elasticity literature [4]. Using the standard conversion formulae taking us between the different standard nonlinear stress definitions, we find that the second Piola-Kirchhoff stress is given by the simple expression

$$S_{ij} = \frac{\delta E}{\delta \epsilon_{ij}}. \quad (33)$$

Inserting the actual energy from Eq. ??(34), we will find

$$\begin{aligned} S_{ij} = & [T_1] \delta_{ij} \\ & - s_1 \left[T_1 - \frac{G_1}{\alpha^2} - \frac{a^2}{\alpha s_1} \tilde{M} \partial_k \alpha \partial_k s_1 \right] \sigma_{ij}^{(3)} \\ & - s_2 \left[T_1 - \frac{G_2}{\alpha^2} - \frac{a^2}{\alpha s_2} \tilde{M} \partial_k \alpha \partial_k s_2 \right] \sigma_{ij}^{(1)}, \end{aligned} \quad (34)$$

where

$$\begin{aligned} T_1 = & -\frac{a^2}{\alpha} \left[\tilde{M} \partial_k \partial_k \alpha + \tilde{M}' \partial_k \alpha \partial_k \alpha \right] - 2 \frac{a^2}{\alpha^2} \tilde{M} \partial_k \alpha \partial_k \alpha \\ & + \frac{1}{2\alpha} \left[\frac{\ell^2}{a^2} M' + a^2 \tilde{M} \partial_k \alpha \partial_k \alpha \right. \\ & \left. + \frac{s_1^2}{2} \left(G_1' - 4 \frac{G_1}{\alpha} \right) + \frac{s_2^2}{2} \left(G_2' - 4 \frac{G_2}{\alpha} \right) \right]. \end{aligned} \quad (35)$$

While the 2nd Piola-Kirchhoff doesn't admit a direct physical interpretation in terms of real traction forces across material surfaces, it is mathematically convenient and can easily be converted to the more physically relevant Cauchy stress with the formula

$$\sigma_{ij} = \frac{1}{\det[F]} F_{ik} S_{kl} F_{jl}. \quad (36)$$

Proposed new SI section explaining the simulation protocol

To probe the viability of boundary control of soft conformal modes, we perform numerical simulations of a simplified version of the

RS lattice. In this case the lattice is composed purely of Hookean springs of identical stiffness k connected at frictionless nodes. As shown in Fig. (6), each rigid square element (grey) is emulated by a grouping of six springs. While a single cross-spring is sufficient to render a square element rigid, the inclusion of both ensures that the spring ensemble will obey the same symmetry properties as the elastic RS metamaterial.

To actuate a particular soft conformal mode, stiff springs are added at the boundary, as shown in Fig. 1. These boundary springs are taken to have stiffness 10^4k , so that they will act as rigid constraints, realizing their rest lengths very accurately compared to the soft mechanics of the bulk material. While the rest lengths in the interior (blue bonds) are chosen to leave the square shape at zero energy, the rest lengths of the boundary springs are varied to match some target local dilation. For three patterns of boundary dilation, we set these rest lengths and identify force balanced states using the conjugate gradient numerical minimization procedure “minimize” from the `scipy.optimize` toolkit in python. At the same time, using the methods from SI Sec. IX, and as illustrated in the main text Fig. 3d&e, predictions for displacements may be generated from this input set of boundary dilations. As shown in main text Fig. 4, these are in good qualitative agreement with the numerically identified force balanced configurations.

Figure 1: **Simple spring model of RS metamaterial and boundary control** The numerical method employed to investigate the viability of boundary control of the RS elastic structure is accomplished using a structure of linear springs (blue lines) connect at frictionless nodes (green dots), approximating a collection of rigid squares (grey regions, displayed for visual convenience). Additional springs are added at the boundary (red lines) and the rest lengths varied to reliably actuate soft modes as shown in the main text Fig.4.

REVIEWERS' COMMENTS

Reviewer #1 (Remarks to the Author):

Overall, the revised version appropriately addresses the points of the referees, and I therefore recommend to accept the manuscript for publication. One point that is still unclear in the manuscript, and I leave it for the authors to handle in the proof stage, is the definition of s_1 and s_2 in Eq. (2). As I mentioned in the first report, I expected the energy to depend on the invariants of C , assuming the medium is isotropic. In the response, the authors explain that the medium is orthotropic, hence cannot depend on the invariants of C alone, with which I agree. However, in the manuscript I did not encounter any details regarding the symmetry of the medium. Furthermore, in finite elasticity, material anisotropy is captured by additional (psuedo) invariants that are functions of C and the preferred directions. I refer the authors to, e.g., the works of Ericksen and Rivlin, Arch.

Ration. Mech. Anal. 3, 281–301. (1954), and Spencer, in Continuum theory of the Mechanics of Fiber-Reinforced Composites, CISM Courses and Lectures, vol. 282. To make connection with the well-established theory of anisotropic materials in nonlinear elasticity, can the authors relate s_1 and s_2 to those psuedo invariants?

Reviewer #2 (Remarks to the Author):

The authors answered my questions in a satisfying way. I recommend it for publication.

Reviewer #3 (Remarks to the Author):

The authors did a great job answering reviewers' questions and updating the manuscript. I recommend the publication of this paper.